# Bat species assemblage predicts coronavirus prevalence

Magdalena Meyer [1,7] ✉, Dominik W. Melville [1,7], Heather J. Baldwin [1,2], Kerstin Wilhelm [1], Evans Ewald Nkrumah[3], Ebenezer K. Badu[3], Samuel Kingsley Oppong[3], Nina Schwensow[1], Adam Stow [2], Peter Vallo[1,4], Victor M. Corman [5,6], Marco Tschapka[1], Christian Drosten [5,6] & Simone Sommer [1] ✉

Anthropogenic disturbances and the subsequent loss of biodiversity are altering species abundances and communities. Since species vary in their pathogen competence, spatio-temporal changes in host assemblages may lead to changes in disease dynamics. We explore how longitudinal changes in bat species assemblages affect the disease dynamics of coronaviruses (CoVs) in more than 2300 cave-dwelling bats captured over two years from five caves in Ghana. This reveals uneven CoV infection patterns between closely related species, with the alpha-CoV 229E-like and SARS-related beta-CoV 2b emerging as multi-host pathogens. Prevalence and infection likelihood for both phylogenetically distinct CoVs is influenced by the abundance of competent species and naïve subadults. Broadly, bat species vary in CoV competence, and highly competent species are more common in less diverse communities, leading to increased CoV prevalence in less diverse bat assemblages. In line with the One Health framework, our work supports the notion that biodiversity conservation may be the most proactive measure to prevent the spread of pathogens with zoonotic potential.

Man-made habitat destruction and climate change in the Anthropocene have launched the 6th major extinction event in our planet's history[1,2]. Simultaneously, newly emerging or re-emerging diseases are becoming more and more frequent[3,4], with chytrid fungal diseases threatening global amphibian diversity[5], rampant mycobacterial-induced tuberculosis spreading among trophic levels[6] and novel zoonotic coronaviruses (CoVs) causing three severe outbreaks in two decades[7–9]. This attests to the inseparable nature of environmental, animal and human health, recognised under the One Health concept[10–13]. How closely these sub-areas are interlinked becomes particularly clear in the case of zoonotic pathogens, i.e., transmitted from animals to humans and vice versa. In fact, 60.3 % of emerging infectious diseases in humans are attributed to zoonoses, of which 71.8 % are estimated to originate in wildlife[10]. Habitat destruction and climate change create more opportunities for pathogens to jump hosts and cause a reshuffling of species assemblages, often resulting in depauperated species communities favouring species resilient to disturbances[12,14–17]. This shift can impact infection dynamics by altering host-pathogen interactions, potentially influencing disease risk and transmission patterns[4,18]. Understanding which precise aspects of biological communities mechanistically determine disease risk is crucial for effective disease prevention and targeted management strategies[18–20].

Although there are roughly 1400 known species of bats globally, their diversity tends to decrease sharply in anthropogenically modified

[1]Institute of Evolutionary Ecology and Conservation Genomics, Ulm University, Ulm, Germany. [2]School of Natural Sciences, Macquarie University, Sydney, New South Wales, Australia. [3]Department of Wildlife and Range Management, Kwame Nkrumah University of Science and Technology, Kumasi, Ghana. [4]Institute of Vertebrate Biology, Czech Academy of Sciences, Brno, Czech Republic. [5]Charité – Universitätsmedizin Berlin Institute of Virology, Berlin, Germany. [6]German Center for Infection Research (DZIF), Berlin, Germany. [7]These authors contributed equally: Magdalena Meyer, Dominik W. Melville. ✉e-mail: magdalena.meyer@uni-ulm.de; simone.sommer@uni-ulm.de

habitats, often favouring a small number of resilient species, which in turn become dominant[21,22]. More than a third of all bat species are considered threatened or data deficient today[22]. Habitat fragmentation, guano mining, bushmeat hunting, and culling in the case of human-bat conflicts are common causes for the decline of bat species[21–25]. This loss of bat species diversity diminishes essential ecological services provided by these animals, such as seed dispersal, pollination, nutrient distribution or insect control[26,27]. Simultaneously, human encroachment into bat niches poses a risk for the spread of pathogens[28,29]. Three of the most recent outbreaks of beta-CoVs have presumably originated in bats[7–9], and two common cold agents caused by alpha-CoVs can also be traced back to a recent ancestor in bats[30,31]. Along with rodents, primates and cetartiodactyls, bats count among the most significant reservoirs of viruses, many of which have zoonotic potential[16,32,33], but the number of viruses found in bats might be in fact proportional to their species richness when compared to other taxonomic groups[34]. Simply put, owing to the focus on bats in viral discovery research, there is a perception that bats somehow host more viruses than other taxa[34,35]. That said, bats have a unique immune system[36–38] with many adaptations linked to flying, possibly responsible for their ability to withstand and recover from infections ('flight as fever' hypothesis[39]), even though viral shedding rates remain high during active infections. As the only mammal capable of powered flight, bats are exceptionally mobile and often congregate in high densities around roosts or food sources, offering opportunities for intra- and interspecies transmission of pathogens over long distances[28,40]. Habitat loss may additionally accentuate crowding[41]. Roosting communities also vary in age composition and seasonally reach high densities of young bats, often naïve to circulating pathogens[42,43]. With this in mind, our study set out to explore which aspects of bat species assemblages predict CoV prevalence and infection likelihood.

Ghana is one of Africa's six bat diversity hotspots[44]. Rapid deforestation, agricultural expansion, and mining threatens the enormous biodiversity of sub-Saharan West Africa[45,46] and the diversity of cave-dwelling bats in particular[23,47,48]. Yet, Ghana remains home to diverse bat communities including roundleaf bats (Hipposideridae[47]). Hipposiderids are known to be speciose with a high cryptic diversity[49–51]. At least four genetically distinct species of the *Hipposideros caffer* complex co-inhabit caves in Ghana[52,53]. Two SARS-related beta-CoVs and the closest relative to the human common cold agent HCoV-229E were previously detected in cave-dwelling Ghanaian roundleaf bats and other co-habiting bats, though differences in susceptibility exist between hipposiderids[30,54,55]. By contrast, the MERS-related beta-CoV 2c strain was only diagnosed in the large-eared slit-faced bat *Nycteris macrotis* in Ghana, possibly indicating a narrow host range of the pathogen[56]. Human encroachment and climate change are predicted to augment the risk of future bat-human spillover in West Africa[12].

In this work, we illustrate the impact of spatio-temporal shifts in bat species assemblages on CoV prevalence and infection likelihood. First, we estimate species diversity at highest taxonomic resolution by deciphering the cryptic *Hipposideros caffer* complex via the mitochondrial *cytb* gene. Second, we calculate the abundance of competent and subadult, i.e., immunologically naïve, hosts as other ecologically relevant pillars determining CoV infection dynamics and merge the data with alpha- and beta-CoV infection information from 2,300 bats. From this the alpha-CoV 229E-like and beta-CoV 2b emerged as multi-host pathogens. However, we demonstrate variation in CoV competence between bat species, with highly competent species being more prevalent in less diverse communities. This results in higher CoV prevalence in bat assemblages with lower species diversity. We argue that by maintaining species diversity as surrogate for important underlying ecological drivers that determine disease spread, effective conservation management can be married with pandemic prevention strategies.

## Results

### Species diversity and spatio-temporal community composition
Over the course of two years, approximately 2,300 bats involved in this study were sampled across five sites in Ghana. Most bat species were clearly identified based on morphological criteria, except for those belonging to the *Hipposideros caffer* complex (Fig. 1A). Additional *cytb* sequencing uncovered the species identity of 1,172 bats belonging to the cryptic *Hipposideros caffer* complex and showed that the abundance and distribution of *Hipposideros* bat species differed across sites (Fig. 1B). *Hipposideros caffer B* was rare and only found in the roosting caves of Buoyem 1 and 2 as well as Forikrom. By contrast, *Hipposideros caffer C* and *D* occurred at all sites. Overall, *Hipposideros caffer D* was most abundant, accounting for 38% of all captured bats (Table 1), followed by the morphologically distinct *Hipposideros abae*, whereas *Rousettus aegyptiacus* was infrequently encountered. The species community composition differed between sampling sites ($F_{4,39} = 6.85$; $R^2 = 0.31$; $p < 0.001$) and periods ($F_{11,39} = 1.99$; $R^2 = 0.25$; $p = 0.019$; Supplementary Data 1 and 2). Furthermore, reproductive cycles could be inferred from the relative abundance of subadults at different time points, which resemble seasonal dynamics (Fig. 1C).

### CoV screening and differences in host competence
Of the 2362 bats that were assigned to 11 species and five families, we recorded 1,113 CoV infections among nine different species and four families (Table 1). These CoV infections split into four CoV clades. While the genomes of alpha-CoV 229E-like and beta-CoV 2c have already been described and analysed in detail[30,54,56], a Bayesian phylogenetic analysis of partial RdRp fragments placed the previously undescribed clades CoV 2b and 2bBasal among SARS-related beta-CoVs (Supplementary Fig. 1).

The alpha-CoV 229E-like was found only in bats of the genus *Hipposideros*, with species belonging to the cryptic *Hipposideros caffer* complex being the most frequently infected. Similarly, the beta-CoV 2b was predominantly found amongst hipposiderids, though few individuals of other genera were also infected. The beta-CoVs 2bBasal and 2c showed markedly fewer cases of cross-species detection.

The most abundant bat species *Hipposideros caffer D* represented a suitable host for three of the four CoVs. In contrast, no CoV infections were detected in the relatively common *Coleura afra*. The results suggest that the alpha-CoV 229E-like and beta-CoV 2b strain are multi-host pathogens, whereas the beta-CoVs 2bBasal and 2c infect fewer host species. The data not only suggested differences in CoV prevalence between bat species but also revealed noticeable spatial and temporal variation (Fig. 1D). The differences in infection prevalence was statistically underscored when analysing infection probability. *Hipposideros caffer B-D*, *Hipposideros abae* and all subadults, irrespective of species, were more likely to be infected with CoV 2b, and *Hipposideros caffer B-D* and all subadults with CoV 229E-like when compared to all other bat species (Supplementary Table 8). But even among hipposiderids differences existed (Supplementary Table 9)[55]. To quantify differences in the hosts' ability to transmit the viruses, we compared viral load across species and age categories, using Ct values as proxy. The Ct value differed between species for both CoVs (Supplementary Fig. 2A and 2B; alpha-CoV 229E-like: $F_{3,310} = 3.23$, $p = 0.023$; beta-CoV 2b: $F_{4,566} = 7.45$, $p < 0.001$). The Ct value of subadults was lower than of adults for both viruses (Supplementary Fig. 2C and 2D; alpha-CoV 229E-like - subadult: $30.61 \pm 4.18$ SD; adult: $32.61 \pm 3.93$ SD, $p < 0.001$; beta-CoV 2b - subadult: mean $31.21 \pm 3.57$ standard deviation; adult: $32.72 \pm 3.44$ SD, $p < 0.001$), suggesting a more acute infection in subadults (Supplementary Fig. 2C and 2D).

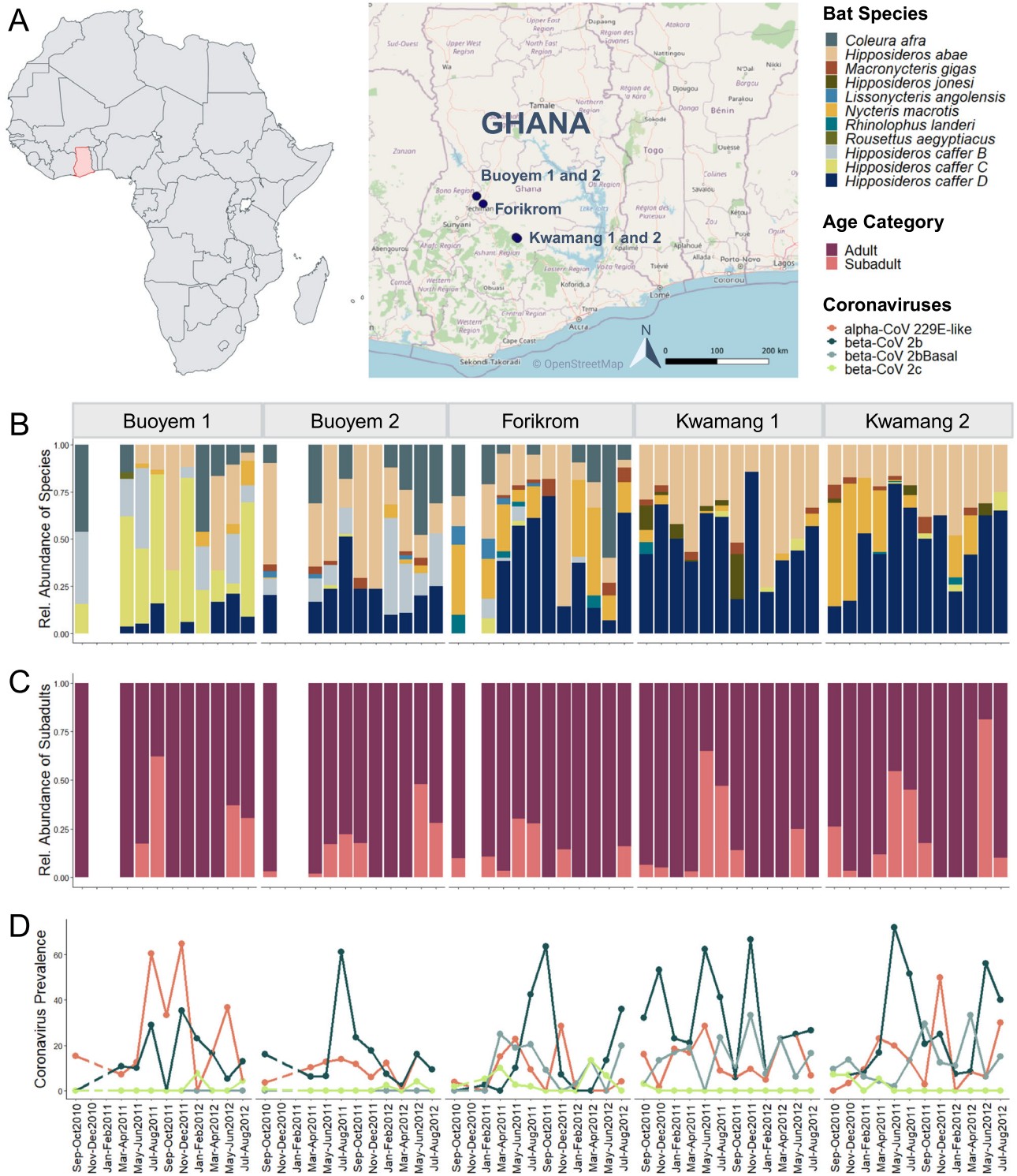

**Fig. 1 | Spatio-temporal community characteristics and coronavirus prevalence.** Sample sites and sampling regime (**A**) over a two-year field study in central Ghana. The relative community composition (**B**), abundance of subadults (**C**) and coronavirus prevalence (**D**) of captured bats at the five bimonthly-sampled caves. Map was created with OpenStreetMap and adapted in BioRender.com. Source data are provided as a Source Data file.

## Correlation between species diversity and CoV prevalence

The prevalence of alpha-CoV 229E-like was weakly but negatively correlated with bat species diversity (Shannon; Fig. 2A; r(55) = −0.33, p = 0.015), while the prevalence of beta-CoV 2b showed a moderate negative correlation with bat species diversity (Shannon; Fig. 2B; r(55) = −0.51, p < 0.001; for Simpson and species richness results see Supplementary Fig. 3A–D). The negative correlation between CoV prevalence and indices that place weight on species abundance and community evenness motivated us to include the relative abundance of susceptible bat species in our follow-up models.

## Community assemblage and CoV infection probability

Lastly, we investigated the influence of community characteristics other than species diversity to explain the individual infection

**Table 1 | Overall species occurrence and coronavirus (CoV) infections amongst bats inhabiting five roosting sites in central Ghana**

| Species | Family | n | CoV 229E-like (%) | CoV 2b (%) | CoV 2bBasal (%) | CoV 2c (%) |
|---|---|---|---|---|---|---|
| *Coleura afra* | Emballonuridae | 151 | 0 (0.00) | 0 (0.00) | 0 (0.00) | 0 (0.00) |
| *Hipposideros abae* | Hipposideridae | 705 | 36 (5.11) | 59 (8.37) | 2 (0.28) | 0 (0.00) |
| *Hipposideros caffer B* | Hipposideridae | 136 | 18 (13.24) | 20 (14.71) | 0 (0.00) | 0 (0.00) |
| *Hipposideros caffer C* | Hipposideridae | 124 | 39 (31.45) | 21 (16.94) | 0 (0.00) | 0 (0.00) |
| *Hipposideros caffer D* | Hipposideridae | 907 | 221 (24.37) | 469 (51.70) | 190 (20.95) | 0 (0.00) |
| *Hipposideros jonesi* | Hipposideridae | 38 | 0 (0.00) | 2 (5.26) | 0 (0.00) | 0 (0.00) |
| *Lissonycteris angolensis* | Pteropodidae | 19 | 0 (0.00) | 1 (5.26) | 0 (0.00) | 0 (0.00) |
| *Macronycteris gigas* | Hipposideridae | 44 | 0 (0.00) | 1 (2.27) | 1 (2.27) | 1 (2.27) |
| *Nycteris macrotis* | Nycteridae | 221 | 0 (0.00) | 1 (0.45) | 0 (0.00) | 30 (13.57) |
| *Rhinolophus landeri* | Rhinolophidae | 15 | 0 (0.00) | 1 (6.67) | 0 (0.00) | 0 (0.00) |
| *Rousettus aegyptiacus* | Pteropodidae | 2 | 0 (0.00) | 0 (0.00) | 0 (0.00) | 0 (0.00) |
| Total | | 2362 | 314 (13.29) | 575 (24.34) | 193 (8.17) | 31 (1.31) |

likelihood. For the alpha-CoV 229E-like, the models that considered Shannon diversity, the relative abundance of each respective species, and the relative abundance of subadults explained between 18% to 62% of the variation in infection probability (Supplementary Data 3).The model explaining the highest variation emphasised the significance of Shannon diversity (Fig. 3A and Table 2, $p = 0.019$), the relative abundance of subadults (Fig. 3E and Table 2, $p = 0.034$) and the relative abundance of *Hipposideros caffer C* (Fig. 3C and Table 2, $p = 0.042$) as key factors influencing alpha-CoV 229E-like infection risk. Conversely, the relative abundances of *Hipposideros abae*, *Hipposideros caffer B* and *D* as well as *Coleura afra* and *Nycteris macrotis* were not associated with infection likelihood (Fig. 3C and Supplementary Data 4). Simpson diversity index (Supplementary Data 5) and species richness (Supplementary Data 6) were not associated with alpha-CoV 229E-like infection risk either.

For the beta-CoV 2b, the models incorporating Shannon diversity, the relative abundances of certain species, and the relative abundances of subadults were able to explain between 27% and 96% of the variation in infection probability (Supplementary Data 3). Shannon species diversity (Fig. 3B and Table 2, $p \leq 0.011$) was negatively associated with beta-CoV 2b infection likelihood. The relative abundances of *Hipposideros abae* and *Nycteris macrotis* also exhibited a negative association with infection likelihood (Fig. 3D and Table 2, $p = 0.003$ and $p = 0.042$). In contrast, the relative abundance of *Hipposideros caffer D* (Fig. 3D and Table 2, $p < 0.001$) and the abundance of subadults (Fig. 3F and Table 2, $p \leq 0.011$) were consistently found to positively influence the likelihood of acquiring an infection. The relative abundances of *Hipposideros caffer B* and *C* as well as *Coleura afra* had no effect on infection risk (Fig. 3C and Supplementary Data 4). Simpson diversity mirrored the results of Shannon (Supplementary Data 5), while species richness was not associated with beta-CoV 2b infection probability in the model with *Hipposideros caffer D* or *Nycteris macrotis* (Supplementary Data 6).

## Discussion

Man-made habitat destruction and climate change cause species extinctions and a reshuffling of species communities. These changes in species assemblages may facilitate the spread and persistence of diseases owing to differences in host competence. Initially, we showcase that differences in bat species assemblages and prevalence of four different CoVs exist among Ghanaian roosting communities. Combining this information, we show a weak to moderate negative relationship between bat species diversity and prevalence of two multi-host CoVs belonging to two different genera (alpha- and beta-CoVs), and that, aside from bat species diversity, the abundance of competent host species and subadults influence CoV infection probability. All in all, we provide evidence that shifts in species communities towards competent hosts in anthropogenically disturbed areas may contribute to the spread and persistence of pathogens in bats.

With more than 100 bat species, the sub-Saharan bat diversity is particularly high[44]. Hipposideridae was the most common bat family encountered over our two-year sampling period. Hipposiderids are exceptionally species rich in both Africa and Asia, but much taxonomic variation remains hidden behind cryptic species structures[49,50,57]. Characterisation of the mitochondrial *cytb* region untangled previously cryptic diversity of Hipposideros species[51], and revealed distinct community compositions of the formerly morphologically indistinguishable *Hipposideros caffer* complex over time and space in Ghana. Differences in echolocation signals[52] and immunogenetic diversity[55] further consolidates the lineage's taxonomic distinction.

Yet another trait that separates the lineages seems to be their competence to resist CoVs. We found uneven infection patterns with alpha- and beta-CoVs among hipposiderids and other bat species. Infections with alpha-CoV 229E-like variants were previously reported from hipposiderids in Ghana[30,54], Kenya[58], Zimbabwe[59], Gabon[60], Mozambique[61], and most recently in Cameroon[62]. Sharing 91.90% sequence identity, the bat CoV 229E-like represents the presumed most recent common ancestor to HCoV-229E[54]. Genomic changes to the open reading frame 8 and deletion of the spike gene in HCoV-229E may have facilitated host transition to humans[30], but equally, host-switching is common in alpha-CoVs[63]. Beta-CoV 2b and 2bBasal phylogenetically group with the SARS-related CoVs[54]. Unlike CoV 2bBasal, which seemed to infect only *Hipposideros caffer D*, CoV 2b infected all Hipposideridae, but was also detected at low prevalence in other bat species. SARS-related CoVs were previously identified from African Rhinolophidae and Hipposideridae[64,65] and rarely Molossidae[66]. We corroborate these findings, and discovered differences in susceptibility and transmission risk between species inferred from Ct values, serving as a proxy for viral load. Such findings might indicate a tie between the diversification of CoV strains and the species-rich families of Hipposideridae and Rhinolophidae[32,67]. Another hypothesis states that speciose roost communities facilitate virus cross-species transmission by promoting RNA-virus recombination during co-infection[68,69]. However, we previously found only the two beta-CoVs 2b and 2bBasal to co-occur frequently, while co-infections between beta-CoV 2b and alpha-CoV 229E-like were less common[55].

In our study, the more competent host species occurred in higher abundances than the less competent hosts in roosting communities facing various anthropogenic disturbances. More than 40% of local survey respondents frequented these bat caves for religious, touristic,

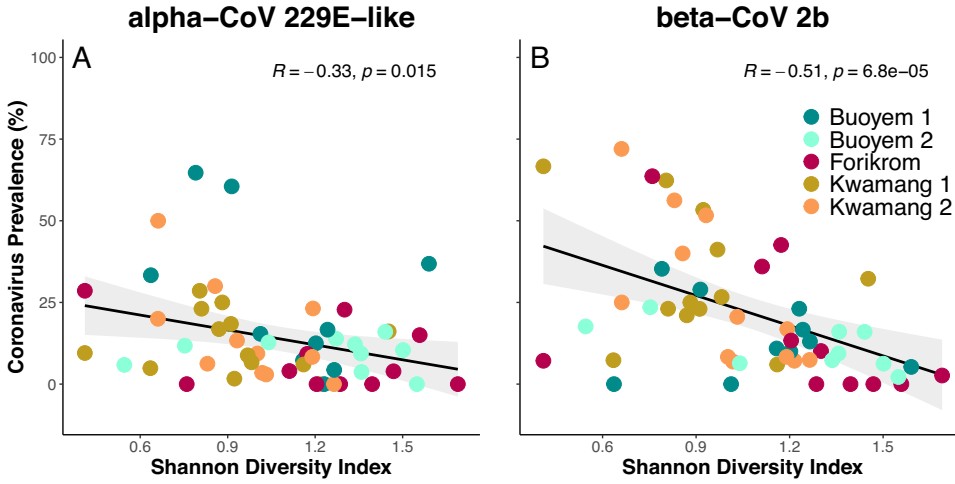

**Fig. 2 | Diversity-disease relationship.** Spearman-correlation between coronavirus prevalence (alpha-CoV 229E-like, beta-CoV 2b) and Shannon species diversity index (**A, B**). Solid line and grey band show best fit and 95 percent confidence interval, respectively. Source data are provided as a Source Data file.

economic or spiritual reasons or simply to fetch water[23]. Additionally, the surrounding matrix is subject to agricultural clearances[47]. Habitat alteration generally is a key driver of biodiversity decline[70], and certain bat species react particularly sensitive to change in pristine vegetation[21,71]. In Ghana too, bat species diversity was lowest in agriculturally disturbed landscapes[72]. The dominance of putative reservoir species for either CoV in the taxonomically less speciose but disturbed communities suggests that these species are more resilient. Our observation aligns with the commonly observed but not fully understood phenomenon that species, which are more resilient to human disturbance, are often competent hosts[16,73], particularly for pathogens with zoonotic potential[4,74]. The picture emerges that less diverse bat assemblages dominated by competent host species experience a higher CoV prevalence and infection likelihood, and such assemblages are more commonly found in habitats disturbed by humans.

A shift in species assemblage towards more competent hosts would explain the negative diversity-disease relationship, a pattern found frequently following human disturbance[18,75]. Such changes in which competent hosts become dominant over non-competent diluter species, is a concept tangential to the dilution effect[76–78]. The dilution effect hypothesis proposes that high biodiversity shields from the spread and persistence of pathogens[18,19,79–81]. In other words, in a genetically homogenous host community pathogens are thought to spread more easily. Two mechanisms are at play: Transmission interference invokes the potential for less competent host species to intercept vector-transmitted disease agents[82], while susceptible host regulation occurs when the presence of non-competent hosts lowers intra-specific transmission of directly transmitting pathogens among the competent host population[83]. The latter can apply to non-vectored disease agents[84–86] like CoVs. However, diversity-disease patterns need to be interpreted in the context of specific host and species assemblages. The impact of changes in biodiversity on disease dynamics will vary depending on the ecological interactions and characteristics of the host community[78,87,88], or put simply, on the manner in which species are introduced to or removed from the community[76]. When correlational studies report the dilution effect, as, for instance, a negative host diversity-virus prevalence relationship in pollinator communities infected with RNA viruses[85], or rodents infected with zoonotic hantaviruses[86], it is often not possible to fully disentangle the effects of changes in host abundance from intrinsic properties of biodiversity, as is the case in our study.

Besides, a negative diversity-disease relationship is just one of many potential outcomes[89–94]. The direction and strength of disease-diversity relationships are highly context-dependent[75,95,96], and this

brought into question whether one can even expect a universal relationship between diversity and disease[20]. In our case, the prevalence of both multi-host pathogens, the alpha-CoV 229E-like and beta-CoV 2b, showed a weak to moderate negative correlation with bat species diversity, which could be interpreted as correlational evidence for the dilution effect hypothesis in a multi-host-multi-pathogen system. However, less diverse assemblages coincided with higher abundance of competent hosts and fewer non-competent hosts. The fact that diversity metrics that place more weight on abundance and evenness were more strongly correlated with CoV infection prevalence than bat species richness per se speaks to the importance of host abundance[84,95]. For instance, rodent and avian hosts density rather than their diversity determined the prevalence of a Hepacivirus[97] and the West Nile Virus[98], respectively, and SARS-related CoVs were most prevalent in a Chinese cave inhabited by multiple bat species when its primary host, the horseshoe bat (*Rhinlophus sinicus*), was more abundant[68]. In our study too, the relative abundance of the most common host species, *Hipposideros caffer D* and less common *Hipposideros caffer C* was positively associated with infection probability with beta-CoV 2b and alpha-CoV 229E-like, respectively. By contrast, *Hipposideros abae* and *Nycteris macrotis* abundances were associated with a lower likelihood of CoV 2b infections, suggesting that their presence reduced transmission risk. This emphasises that the direction of the diversity-disease relationship will depend on the spatiotemporally variable composition of the species assemblage[20,99].

In addition, the (immuno-)genetic diversity or other ecological features, such as reproductive bouts, are thought to underly disease dynamics. The immunogenetic profile of *Hipposideros caffer C* and *D* vary when compared with *Hipposideros abae*. Whereas *Hipposideros caffer C* and *D* comprised major histocompatibility complex (MHC) class II supertypes associated with susceptibility to CoV 229E-like and 2b, *Hipposideros abae* lacks an MHC supertype-CoV 2b association[55]. Moreover, bat roosts with a high abundance of immature individuals appear frequently as hotspots for infections[42,43]. Accordingly, the infection likelihood of both multi-host CoVs was partly determined by the relative abundance of subadult bats, although the precise mechanisms remain elusive and could be due to a combination of behavioural changes, physiological stress and waning benefits of maternal immunity[100]. Nevertheless, we found a lower Ct value in subadults when compared with adults. Low Ct values imply a high viral load and suggest a more acute infection and likelihood for transmission. The relative instability of host immunity and gut microbiome-mediated defences at younger age could be key to understand such recurring findings[101,102].

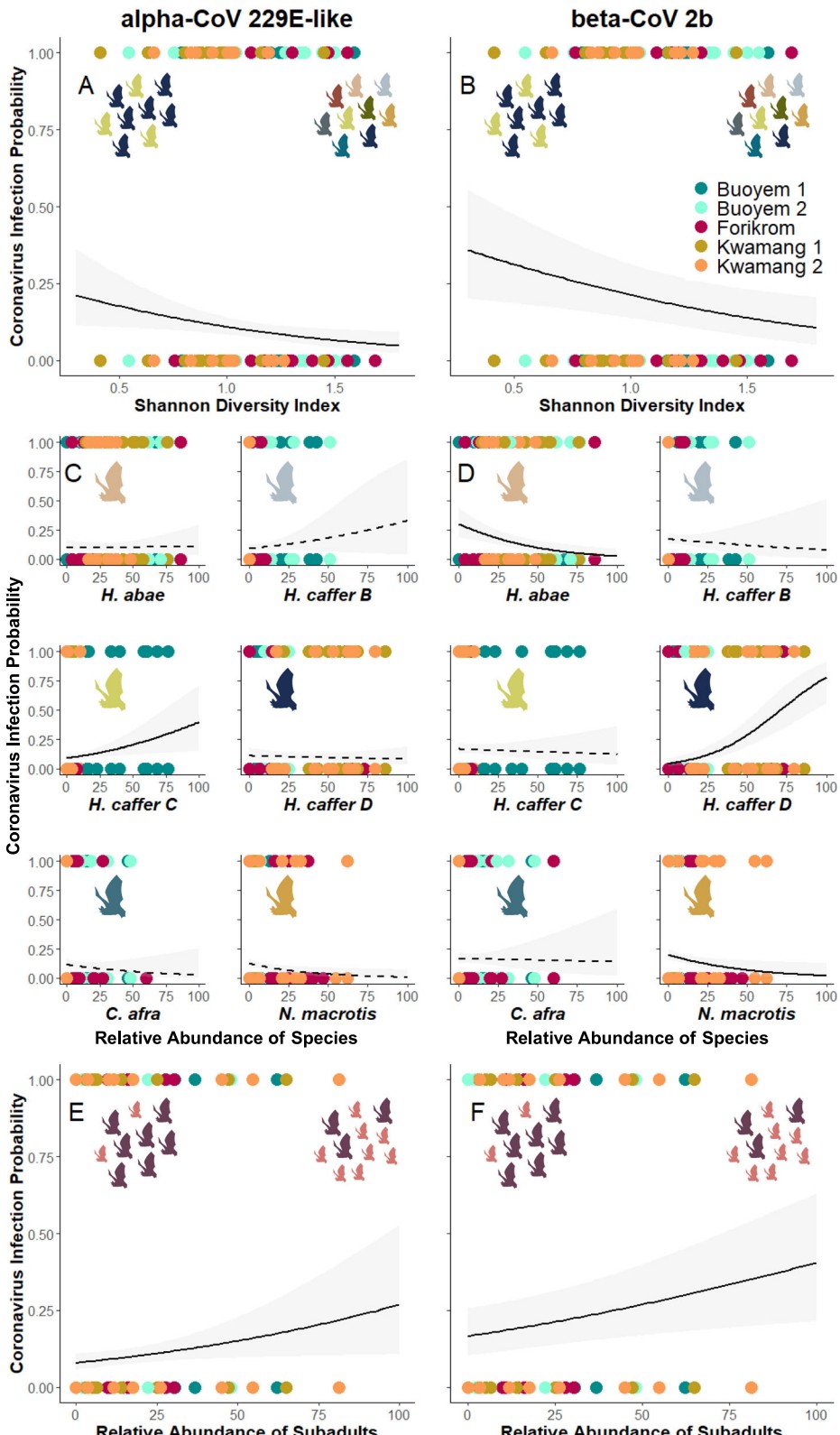

**Fig. 3 | Ecological determinants of coronavirus infection likelihood.**
Coronavirus (CoV) infection probability (alpha-CoV 229E-like, beta-CoV 2b) in relation to the Shannon Diversity Index (**A, B**), the relative abundance of the most common (*n* < 50) species *Hipposideros (H.) abae* (apricot), *H. caffer B* (light blue), *C* (yellow), *D* (dark blue) as well as *Coleura (C.) afra* (slate grey) and *Nycteris (N.) macrotis* (orange) (**C, D**), and the relative abundance of subadults (adults=magenta; subadults=pink) (**E, F**) in each of the five sampling sites in central Ghana modelled using generalised linear mixed effect models. Solid and dashed lines represent significant (FDR-corrected) and non-significant fitted model curves, respectively, and lightly shaded area the respective 95% confidence interval. Icons were created with BioRender.com. Source data are provided as a Source Data file.

**Table 2 | Results from the generalised linear mixed effect models comparing the effect of Shannon Diversity Index, the relative abundance of *Hipposideros (H.) abae*, *H. caffer C* and *D* and *Nycteris (N.) macrotis* as well as subadults on coronavirus (CoV) infection probability**

| | Predictors | Odds Ratios | CI | p-value | p-value adj. |
|---|---|---|---|---|---|
| **alpha-CoV 229E-like** | *CoV 229E-like ~ Shannon + Rel. Abundance of H. caffer C + Rel. Abundance of subadults + (1|Site/Sampling time)* | | | | |
| | (Intercept) | 0.27 | 0.09 – 0.77 | **0.014** | |
| | Shannon Diversity Index | 0.32 | 0.13 – 0.80 | **0.014** | **0.019** |
| | Rel. Abundance of *H. caffer C* | 1.02 | 1.01 – 1.03 | **0.007** | **0.042** |
| | Rel. Abundance of subadults | 1.01 | 1.00 – 1.03 | **0.028** | **0.034** |
| **beta-CoV 2b** | *CoV 2b ~ Shannon + Rel. Abundance of H. abae + Rel. Abundance of subadults + (1|Site/Sampling time)* | | | | |
| | (Intercept) | 7.51 | 1.85 – 30.51 | **0.005** | |
| | Shannon Diversity Index | 0.05 | 0.02 – 0.14 | **<0.001** | **<0.001** |
| | Rel. Abundance of *H. abae* | 0.97 | 0.96 – 0.99 | **0.001** | **0.003** |
| | Rel. Abundance of subadults | 1.02 | 1.01 – 1.03 | **0.001** | **0.001** |
| | *CoV 2b ~ Shannon + Rel. Abundance of H. caffer D + Rel. Abundance of subadults + (1|Site/Sampling time)* | | | | |
| | (Intercept) | 0.12 | 0.03 – 0.40 | **0.001** | |
| | Shannon Diversity Index | 0.36 | 0.16 – 0.79 | **0.011** | **0.011** |
| | Rel. Abundance of *H. caffer D* | 1.04 | 1.03 – 1.06 | **<0.001** | **<0.001** |
| | Rel. Abundance of subadults | 1.01 | 1.00 – 1.02 | **0.011** | **0.011** |
| | *CoV 2b ~ Shannon + Rel. Abundance of N. macrotis + Rel. Abundance of subadults + (1|Site/Sampling time)* | | | | |
| | (Intercept) | 1.57 | 0.56 – 4.42 | 0.395 | |
| | Shannon Diversity Index | 0.13 | 0.05 – 0.34 | **<0.001** | **<0.001** |
| | Rel. Abundance of *N. macrotis* | 0.98 | 0.96 – 1.00 | **0.021** | **0.042** |
| | Rel. Abundance of subadults | 1.02 | 1.01 – 1.04 | **0.001** | **0.001** |

Significant results are in bold and are presented as original and FDR-corrected p-values.

Taken together, changes in community assemblages in disturbed habitats play a significant role in shaping diversity-disease relationships. While diversity-disease relationships provide valuable insights, they are not universally applicable and may not fully capture ecological intricacies. To better understand the connections between biodiversity loss, wildlife communities, and infectious disease dynamics, further research should consider specific characteristics and interactions within host assemblages, such as immunogenetic or age-related differences, but high taxonomic resolution is certainly key[35]. Given the high zoonotic threat of CoVs generally, and SARS-related CoVs and HCoV-229E-like variants in particular, as well as the ongoing cave utilisation, bat consumption and habitat alterations, spillover events are a serious public health risk in Sub-Saharan Africa. In order to relax this risk, we need to preserve pristine and restore disturbed ecosystems to support a wide range of species and re-establish ecological processes that act as barriers to the spread of disease[103]. An important first step to slow and reverse human encroachment is educating the public about the services provided by animals in an intact ecosystem[26,27,104], followed by community-driven conservation initiatives working towards habitat restoration. However, the effectiveness of these initiatives is inherently tied to the ability to address concurrent socio-economic and cultural needs[105–107].

In summary, shifts in bat community assemblages determine CoV prevalence in disturbed cave sites in central Ghana. We emphasise that the abundance of competent bat species and subadults are key ecological drivers of CoV infection likelihood. Mitigating the risk of future disease spillover from bats to humans has to start with the protection of habitats and wildlife according to a holistic One Health concept.

## Methods

### Study area and sample collection

Research (A04957) and ethics permit (CHRPE49/09/CITES) were granted by the Wildlife Division of the Forestry Commission of the Ministry of Lands, Forestry and Mines. As part of a large longitudinal study (see Supplementary Methods), bats were captured from five caves in central Ghana, West Africa, between August 2010 to August 2012. The sites Buoyem 1 (N7°72′35.833″ W1°98′79.167), Buoyem 2 (N7°72′38.056″ W1°99′26.389), Forikrom (N7°58′97.5″ W1°87′30.299), Kwamang 1 (N6°58′0.001″ W1°16′0.001) and Kwamang 2 (N7°43′24.899″ W1°59′16.501) were sampled every two months (except for two occasions in Buoyem 1 and 2 and one in Forikrom; Fig. 1, Supplementary Table 1). Available information on cave features and anthropogenic disturbance are summarised in the Supplementary Methods and Supplementary Tables 2, 3 and 4.

Bats were captured at each roosting cave site for two nights using mist nets strung along the cave entrances one hour after dusk until dawn. To minimize disturbance to the bats, sampling was paused for a night between the first and second sampling event. Upon capture, species were identified based on morphological characteristics if possible. Animals with distinctly ossified phalangeal joints were categorized as adult, all younger, volant but non-adult individuals as subadult. To account for the possibility of recaptures, animals were tagged with a numbered metal ring (I.Ö. Mekaniska AB, Sweden). In order to identify cryptic species, minimally invasive wing punches (2 mm) were collected and stored in 90% ethanol at −20 °C until further processing. Additionally, to screen the bats for possible infection with CoVs replicating enterically[108], faecal samples were collected and stored in RNAlater (Qiagen, Germany) at −80 °C.

### Species assignment and *cytochrome b* sequencing

Out of 2362 bats captured at cave entrances, species identity of 1172 bats could not be assigned based on morphological characteristics alone, as they belonged to the cryptic *Hipposideros caffer* species complex. We assigned them to one of three lineages (i.e. B, C, and D) proposed to embody distinct species via mtDNA *cytochrome b (cytb)* genotyping[51–53]. The species nomenclature of the *Hipposideros caffer* complex remains unresolved throughout the Afrotropics. Here we use *Hipposideros caffer B*, *C*, and *D* as interim species names. DNA was extracted from wing punch tissue using an ammonium acetate protocol. The mitochondrial *cytb* gene was amplified by PCR using

primers (ID L15225 and H15344, sequences see Supplementary Table 5), previously developed for Sanger sequencing[51] but here adjusted for Illumina sequencing. Usage of specific adapters (Fluidigm, USA) and dual 10 bp barcodes allowed pooling of samples for high-throughput Illumina sequencing. Assignment of species was completed in Geneious 11.1.5 (https://www.geneious.com) using the MAFFT alignment tool.

## Virus screening

To detect different alpha- and beta-CoVs, RNA was purified from faecal material solved in RNAlater stabilization solution using the MagNA Pure 96 system (Roche, Penzberg, Germany). RNA was analysed by real-time reverse transcriptase-PCR. The bat faecal samples were tested for four different CoV clades (Supplementary Methods for more details and Supplementary Table 6 for primer information). The clades include the alpha-CoV strain 229E-like, which is related to the human common cold agent HCoV-229E[30], the SARS-related beta-CoV 2b[54], its variant 2bBasal and the MERS-like beta-CoV 2c[56] that was differentiated by clade-specific real-time PCR. Bats were considered infected if fragments of viral RNA from any of the four virus clades were detected. We only considered samples with Ct values of 38.0 or less to be CoV positive (equivalent to >15 CoV-RNA copies/μL). Please see Supplementary Methods for more details on PCR setup and controls. CoV prevalence was estimated as the proportion of bats with detectable viral RNA in faeces per site and sampling period.

## Statistical analyses

All analyses were performed in the software R (v4.1.1; R Core Team 2022[109]). Species richness, Shannon's diversity index, Simpson's reciprocal index and species-specific as well as subadult abundance were calculated for each site and sampling period based on captures (n = 2362 bats; Supplementary Data 1 and 2, Supplementary Table 7) using the 'vegan' package. Differences in bat species community between sampling sites and periods were assessed using the *adonis2()* function in the 'vegan' package. Virus prevalence was calculated for each bat species, each cave and at each sampling time point.

First, to investigate whether differences in competence exist between species and age categories (subadults, adults) we computed a generalised linear model on individual CoV infection probability as binomial response variable (positive/negative coded as 1/0). Additionally, to assess differences between hipposiderids or whether to regard them as a unilaterally susceptible host group, another generalised linear model was applied with *Hipposideros* species identity as the explanatory variable. As indication as to whether viral load differed between species and age categories, we performed an analysis of variance on the Ct value of CoV positive bats using species and a t-test using age as explanatory variable.

We tested for a correlation between CoV prevalence and species diversity by Spearman correlation, which avoids assumptions about the underlying data distribution and the linearity of the relationship between variables[110], using the *cor.test()* function in the R base package 'stats'. Given a recapture rate of 0.25% in core sites, each sampling site and period were treated as independent sample, which meant no correction was necessary to account for five missing sampling events (Fig. 1, Supplementary Data 1 and 2).

Finally, we applied generalised linear mixed effects models (GLMMs; *glmer()* function in the 'lme4' package) to fit individual CoV infection probability as response variable with either one of the diversity indices (species richness, Shannon or Simpson Index), either of the relative abundance of common bat species (>50 observations; i.e. *Hipposideros abae, Hipposideros caffer B, C* and *D, Coleura afra* and *Nycteris macrotis*) as well as the relative abundance of subadult bats as explanatory variables and sampling period nested within the sampling site as random effects. We used the *dredge()* function in the 'MuMIn' package[111] to identify competitive models based on the Akaike's

adjusted Information Criterion (AICc)[112]. The full model was either the best or fell within a ΔAICc ≤ 2.00. To maximise comparability, we report the dredge summary for all equally competitive models (Supplementary Data 3). For the sake of comparability, the results in the main text, summary tables and visuals report the results from the full model containing all explanatory variables (if the full model was competitive, which was always the case). Additionally, model averaging was performed for all models with ΔAICC ≤ 2.00 (see Supplementary Data 7–9), but never changed the interpretation of the results. The models were checked for multicollinearity using the *check_collinearity()* function in the 'performance' package[113]. Due to the presence of moderate (VIF factor 5-10) to high (VIF factor ≥ 10) multicollinearity among the diversity indices and species abundances, models were run separately for each highly correlated predictor. In all competitive 36 full models, the predictor variables exhibited low correlation (VIF factor below 5). We applied a false discovery rate (FDR) correction using the *p.adjust()* function in the 'stats' package to account for multiple testing and report FDR-corrected p-values for all GLMMs throughout the results section, with a significance level of alpha = 0.05.

## Reporting summary

Further information on research design is available in the Nature Portfolio Reporting Summary linked to this article.

## Data availability

The entire data generated in this study have been deposited on GitHub (https://github.com/MagdalenaMeyer/Bat-species-assemblage-predicts-CoV-prevalence) and figshare (https://doi.org/10.6084/m9.figshare.21982592). Additional data generated in this study are provided in the Supplementary Information. The viral sequencing data used in this study are available in the Genbank database under the following accession codes: HQ166910.1, MT586852.1, MG000872.1, JX869059.2, KJ477102.1, MT084071.1, NC_045512, AY572034.1, NC_004718.3, JX174638.1, JQ410000.1, NC_002645.1, OR482956, OR482957, OR482958, OR482959, OR482960, OR482961, OR482962, OR482963, OR482964, OR482965, and OR482966. Source data are provided with this paper.

## Code availability

The R code for the current study is publicly on GitHub (https://github.com/MagdalenaMeyer/Bat-species-assemblage-predicts-CoV-prevalence) and figshare (https://doi.org/10.6084/m9.figshare.21982592).

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

## Acknowledgements

The research was funded by the German Research Foundation (DFG SO 428/17-1, with samples and metadata obtained by DR 772/3-1 & 7-1, KA1241/18-1, TH 1420/1-1). We thank the chiefs and community leaders of Buoyem, Forikrom and Kwamang for their support, Tatiana Tilley and Thomas Link-Hessing for a helping hand in the lab, Amanda Vicente-Santos for constructive feedback, and Lena Bayer-Wilfert for analytical advice.

## Author contributions

M.M., D.W.M. and S.S. conceived the idea behind the present study. H.J.B. and M.T. organized the field work. E.E.N., E.K.B., S.K.O., P.V., M.T. and H.J.B. collected and archived samples. K.W., D.W.M. and M.M. completed lineage assignment based on previous work from H.J.B., A.S. and P.V. H.J.B., V.M.C. and C.D. generated the viral infection data. M.M., D.W.M. and K.W. analysed data. S.S. and N.S. acquired funding. M.M., D.W.M., and S.S. wrote the first manuscript draft. All authors contributed to the final version of the manuscript.

## Funding

## Competing interests

The authors declare no competing interests.

## Ethical approval

All applicable institutional and/or national guidelines for the care and use of animals were followed.
