## [Peer Review File · Nature Communications]

Bat species assemblage predicts coronavirus prevalenceEditorial Note: This manuscript has been previously reviewed at another journal that is not operating a transparent peer review scheme. This document only contains reviewer comments and rebuttal letters for versions considered at *Nature Communications*.

REVIEWER COMMENTS

Reviewer #2 (Remarks to the Author):

The manuscript submitted by Dr. Meyer and co-authors (manuscript number NCOMMS-23-22117-T) presents the results of a longitudinal study conducted in Ghana over two years. The study focused on surveying bats roosting in caves and examining the presence of alpha- and beta-coronaviruses (CoVs). A total of over 14,000 bat specimens were sampled across 17 sites, but the study focuses on 2362 specimens surveyed on five cave sites. The bat species were identified using external morphology or cytochrome b (cytb) sequencing (for cryptic species), and four CoVs were screened using real-time RT-PCR. CoV prevalence was calculated for each cave at each sampling time point.

To assess the relationship between CoV prevalence and bat species diversity, various diversity and species richness indexes were calculated and tested for correlation. The objective was to evaluate whether the predictions of the dilution effect hypothesis, which suggests higher disease prevalence in less biologically diverse species assemblages, applied to the studied system. Additionally, generalized linear mixed effects models (GLMM) were employed to fit CoV infection probability of the two multi-host pathogens (alpha-CoV 229E-like, and beta-CoV 2b) using the inferred diversity indexes, the relative abundance of the most common and susceptible bat species (*Hipposideros caffer* complex and *H. abae*), and the relative abundance of subadult bats as explanatory variables.

The authors' main conclusions are that there is a connection between bat species diversity and CoV prevalence and that the abundance of the most common bat species and subadults influences the likelihood of CoV infection.

Overall, the manuscript is well-written and well-organized. It presents a detailed account of the study's technical aspects. The analytical approach is appropriate, with a clear

presentation of statistical tests, and the conclusions are supported by the results (but see comments below). The manuscript contributes relevant and original findings, as there have been no prior longitudinal studies involving multi-host viruses in multiple independent bat assemblages. This represents an interesting endeavor to analyze the relationship between diversity and disease in bat coronaviruses.

Major comments:

The main point of this study is to test the predictions of the dilution effect hypothesis (DE) in assemblages of cave-dwelling bats. The problem is that the effects of host community composition or the presence of a highly competent species for the pathogen in question are not isolable under the reported results. The authors show that there is a moderate/low negative correlation between CoV prevalence and species diversity. But there is a specific case under study and not all possible cases, and this makes that the decrease in diversity cannot be isolated from the increase in *H. caffer* relative abundance. So, is it diversity, or is it just *H. caffer*, which is the confirmed reservoir for both multi-host CoVs? This has to be problematized since the results reported in this study can be more simply interpreted as in a previous study involving mixed-species roosts in Yunnan province, China, where a greater prevalence of SARSr-CoVs was found when *Rhinolophus sinicus*, a primary host of SARSr-CoVs, was more abundant in the roost than other species (<https://doi.org/10.1371/journal.ppat.1006698>). Thus, the presence of key species (and their density) included in an assemblage may determine pathogen prevalence and not any intrinsic property of biodiversity (<https://doi.org/10.1890/07-1047.1>).

I suggest reorienting the manuscript in terms of the relative abundance of the *H. caffer* complex. The only aspect that could be at some extent attributable to the DE is the fact that *H. abae*, although being also a potential reservoir for both alpha-CoV 229E-like, and beta-CoV 2b, could function as a less-competent host than *H. caffer*, so when it increases its relative proportion in a given assemblage, the overall prevalence tends to decrease. I suggest the authors discuss this interesting result. But again, it is not about species diversity, but instead of which of these two species is more abundant. I think that the authors should test for other species (individually and grouped as a whole) in the GLMM (and also in

correlation analyses) that are not competent hosts. The species *Coleura afra* is likely a candidate of a non-competent host that when it increases its relative abundance, it produces a decrease in CoV prevalence.

However, the authors should discuss the fact that the effects of host community composition or the presence of a highly competent species for the pathogen in question are difficult to isolate (<https://doi.org/10.1017/s0031182012000200>).

As Keesing et al. (2010) point out:

“For example, if those host species most responsible for amplifying the pathogen tend to persist or even thrive as biodiversity is lost, then disease risk will consistently increase as biodiversity declines. On the other hand, if amplifying species tend to disappear as biodiversity declines, then biodiversity loss will tend to reduce disease risk. These hypothetical possibilities indicate the importance of understanding both the non-random sequences by which species are lost from communities, and whether the species that tend to occur only in more species-rich communities tend to amplify or buffer pathogen transmission”.

In this sense, there is no single rule for disease-diversity relationships. The authors claim that more generalist and susceptible hosts are more prone to prevail in the face of biodiversity loss, but this is not the only possible outcome.

In addition, not any modification in the composition of a given community would have the same outcome. The extent to which mechanisms that are affected by species composition relate to biodiversity would depend on how species are added to (or lost from) a community. These assemblies can be substitutive (individuals of a new species replace individuals of existing species), additive (adding new species adds more individuals to a community), or a combination of both. This can only be assessed by estimating absolute species abundances and the authors work with relative abundances. I suggest the authors discuss this aspect in the manuscript.

For the above-mentioned reasons, it is not appropriate to state that this study provides evidence for the dilution effect hypothesis. In the best case, the results are compatible with the DE hypothesis, but it should be remarked that this is purely speculative since there are

no counterexamples to prove that there is any intrinsic property of biodiversity operating in the studied system. Therefore, I suggest restricting the interpretation to the abundance of the main reservoir (and to the proportion of subadult specimens) and extending the analysis to non-competent/less-competent species.

Minor comments:

Line 87: remove “this diversity-disease relationship”.

Line 90: the DE is not only criticized for its limitations but also its exceptions. The DE is not the only possible outcome of biodiversity loss: no relationship, a context-dependent relationship or amplification effects have been reported in the same manner as DE (see <https://doi.org/10.1371/journal.ppat.1009637>, <https://doi.org/10.1038/s41559-019-1060-6>, and references therein). The authors should present a complete panorama of this topic.

Line 189: the reference cited does not apply qRT-PCR, so it is impossible that the authors can take a Ct from this paper. Please be more cautious and responsible in the use of bibliographic references.

Line 191: the authors should state which is the universe in which the statistical analyses are based (the set of 2362 bats).

Line 209: I suggest that the authors run the same analysis taking the H. caffer complex as a whole, and test if this variable produces stronger correlations and more explanatory models.

Line 259: remove “strong”. If it is moderate, it is not strong.

Line 304: a weak/moderate negative correlation.

Lines 334-336: revise taking into consideration the major comments made above. The authors cannot separate the loss in diversity from the increase in the frequency of H. caffer.

Lines 337-338: the ‘dead end’ explanation is difficult to invoke in pathogens capable of spreading enormous quantities of infective stages into the environment, such as coronaviruses. CoVs are likely to be transmitted passively in co-roosting bats. Long periods of viral shedding in feces, superspreading, and aerosolization, are some of the characteristics that make viral sharing more likely in cave-dwelling bats, independent of

their competence (see <https://doi.org/10.3390/d9030035>).

Lines 340-343: again, the authors analyze only one possible outcome. It could equally happen that more competent hosts disappear in species-depleted communities, or that the change is additive, substitutive, or a combination of both.

Lines 347-348: revise taking into consideration the major comments made above. The authors cannot separate the loss in diversity from the increase in the frequency of *H. caffer*.

Line 349: metrics, not metrices.

Lines 349-351: at least for bat CoV 229-E-like it is obvious that metrics that place more weight on host abundance and evenness will have a stronger correlation with CoV prevalence since the only intervening variable found is the abundance of *H. caffer*. Again, it is not diversity, it is the abundance of *H. caffer*, the most competent reservoir for these CoVs. The counterexample to prove that it would be a matter of species diversity would be a cave where species diversity is diminished but the dominant species is a less-competent host. In this sense, the authors can interpret the case of *H. abae*, as a less-competent host, that when increases its density, at the expense of *H. caffer*, induces a decrease in beta-CoV 2b prevalence. But again, this is not a property of biodiversity, but of these two key species in particular.

Line 383: It is not less diverse bat assemblages, but those in which the natural reservoirs (*Hipposideros caffer* complex and *H. abae*) are abundant.

Reviewer #4 (Remarks to the Author):

Meyer & Schmid et al. explored the relationship between bat species diversity and coronavirus prevalence / infection likelihood among cave-dwelling bat communities in Ghana. The authors contextualise the study as providing empirical evidence for the “dilution effect” (low disease prevalence in biologically more diverse species assemblages).

I want to commend the authors for the wealth of data they have generated on this system. Longitudinal datasets on bat CoVs are scarce, and the authors present a truly impressive sample set that will generate extremely valuable insights into bat-CoV dynamics. However, Reviewers 2 and 3 have raised several points about the study’s inability to investigate the dilution effect, which I agree with, and which have not been adequately addressed by the

authors. I elaborate below, and also give my own separate comments. I hope the authors take the feedback as positive and constructive, as intended.

General Comments Relating to Previous Reviews:

As highlighted by Reviewers 2 and 3, it is likely that key species abundance – and not assembly diversity – may correlate with CoV prevalence. The authors rebuttals against this comment seem to have missed the point. Reviewer 2 did not state that coronavirus-bat relationships are an unsuitable study system to investigate disease-diversity relationships, only that there needs to be at least one counterexample where prevalence increases as species diversity decreases *and where the most abundant species are different from natural reservoirs* to demonstrate the dilution effect. If this situation is not found, then observed patterns cannot be inferred as the result of changes in biodiversity, because patterns may simply reflect changes to key species abundance.

Consider two hypothetical cave roosts, both with two species each because of high disturbance. In cave one the most abundant species is a generalist and CoV reservoir, in cave two the most abundant species is a generalist but NOT a CoV reservoir. Both caves have the same assembly diversity, but the prevalence of CoVs will be higher in cave one where the abundant species is a CoV reservoir. This hypothetical scenario demonstrates how prevalence patterns can reflect key species abundance, and not changes in assembly diversity as predicted by the dilution effect. This phenomenon is described more succinctly by Reviewer 2: “If those host species most responsible for amplifying the pathogen tend to persist or even thrive as biodiversity is lost, then disease risk will consistently increase as biodiversity declines. On the other hand, if amplifying species tend to disappear as biodiversity declines, then biodiversity loss will tend to reduce disease risk. These hypothetical possibilities indicate the importance of understanding both the non-random sequences by which species are lost from communities, and whether the species that tend to occur only in more species-rich communities tend to amplify or buffer pathogen transmission”. “Thus, the presence of key species (and their density) included in an assemblage may determine pathogen prevalence and not any intrinsic property of biodiversity.” (<https://doi.org/10.1890/07-1047.1>).

Looking at figure 1B, the relative abundance of species in the majority of time points, are dominated by Hipposideridae. I see only one time point (of 55) where the *Hipposideros caffer* complex is absent (though where *Hipposideros abae* remain). At this time point, there are 5 species present (comparable to other time points), and negligible CoV prevalence of all clades. This doesn't meet the situation as outlined by Reviewer 2, and could actually indicate that the relative abundance of *Hipposideros* spp. drive infection, not assembly diversity.

Reviewer 2 highlighted that both the *Hipposideros caffer* complex and *Hipposideros abae* are likely natural reservoirs for the two CoV clades analysed, and suggested that the authors analyse the effect of these hosts as a single set. I agree with this comment, and am surprised that the authors didn't accommodate this seemingly straightforward suggestion. The two CoV clades analysed are predominantly associated with hipposiderid and rhinolophid bats (e.g., Anthony et al. 2017, doi: 10.1093/ve/vex012). This makes it difficult to interpret what the diversity metrics mean in relation to the "dilution effect". E.g., compare two hypothetical groups each with 6 species, but one group is comprised entirely of hipposiderid species, and the other with 1 hipposiderid species and 5 non-hipposiderid species. The two groups should have the same diversity index, but vastly different infection potentials for these CoV clades. The authors have attempted to account for the presence of susceptible species, by including the relative abundance of four key hipposiderid species in GLMMs. However, these species are considered in isolation, so that each model set accounts for the relative abundance of one dominant/susceptible species, but not the others. It is therefore still difficult to interpret what the diversity metrics mean in relation to the "dilution effect". I would recommend that the authors first test to see whether infection of individual bats is predicted by species in this dataset (in particular, test this for the hipposiderid – and maybe rhinolophid – subset, to test for differences between these key species). If infection probability is not different between these species, I would consider them as a single dominant host "group" in the GLMM set of models, for the purpose of testing the "dilution effect", as suggested by Reviewer 2.

The authors discuss in the rebuttal that the two modelled CoV clades are found in multiple hosts, even outside the *Hipposideros* complex. I think this is the point (that the CoV clades

are potentially associated with multiple species in the dataset, therefore accounting for the relative abundance of any one species in isolation doesn't capture the whole picture). It would be valuable to include information about CoV clade*bat family/species relationships in the text, to better contextualise the specific CoV-bat species relationships included in the manuscript. This is currently missing from the manuscript. I would also encourage the authors to present a tabular breakdown of species captured per session*site in the SI, so that the readers might be able to interpret species compositions relative to the family groupings. It would also be easier to interpret the species assemblages in Figure 1B if key the susceptible bats (hipposiderid bats) were ordered together at the bottom of bands, and distinctly coloured (e.g., hipposiderid species as shades of red, and non-hipposiderid species as shades of grey).

Additionally, in the presented context of disturbance and biodiversity loss, one might interpret hipposiderid bats to be dominant, generalist species, given their high representation in the data. However, it is my impression that African hipposiderid bats are more specialist species that are sensitive to disturbance (e.g., Webala et al. 2004 <https://doi.org/10.1111/j.1365-2028.2004.00505.x>; Wechuli et al 2016 <https://doi.org/10.1111/aje.12376>). This is relevant to the points raised by Reviewers 2 and 3, because one might expect that with disturbance and loss of hipposiderid bats, prevalence of these specific CoV clades should decrease, regardless of biodiversity. If the authors still want to keep the dilution effect narrative, it could be worth clarifying the potential role of hipposiderid bats as a study system for understanding the dilution effect in the context of disturbance and species composition change. It would also be worth presenting some information on the extent of disturbance at sampled sites, within the methods, so that the reader might interpret to what extent this system represents a disturbed habitat.

Lastly, I would note that Reviewer 2 referred to the vector-borne pathogen literature by way of illustrating the two mechanisms that can generate the dilution effect (transmission interference and susceptible host regulation). These mechanisms (or certainly the second mechanism) can operate for non-vector disease agents, but the point is that it's context specific and hard to prove empirically. The authors rebut "The concept proposes that the presence of a more diverse array of hosts in a community can reduce the chances of an

infected individual encountering a susceptible host, thereby limiting the transmission of disease.” This is true, but one must be able to differentiate between the effects of key species abundance and changes in biodiversity, as described above.

Overall, I think the bat-CoV dynamics in this dataset are interesting and worthy of publication. I would just reframe the narrative of the paper to remove the focus on testing the “dilution effect” and contextualise the changes as they relate to specific species compositions (host vs non-host). The authors could use this as an opportunity to discuss why the dilution effect is not immediately useful as a generalisable concept (in the sense that disease-diversity patterns must be interpreted with regard to specific host/species assemblages).

General Comments, New:

I would encourage the authors to check the relevance and accuracy of all citations. There are several citations that are either unrelated to the statement its supporting, or directly counter the statement – e.g, L81-82: “It is estimated that wild animals are the origin of at least 60 % of zoonotic infections in humans” Allen et al. 2017 (doi:10.1038/s41467-017-00923-8) doesn’t give a statistic of this nature. It shows the spatial distribution and predictors of zoonoses from wildlife; L112-114: “As the only mammal capable of powered flight, bats are exceptionally mobile and often congregate in roosting caves offering opportunities for intra- and interspecies transmission of pathogens over long distances.” Neither reference provided – Letko et al. 2020 (doi:10.1038/s41579-020-0394-z) nor Li et al. 2005 (doi:10.1126/science.1118391) – discuss migration, aggregation, or intra/interspecies transmission. L126-127: “Besides, hipposiderids count amongst the most important reservoirs of CoVs in the paleotropics” Letko et al. 2020 (doi:10.1038/s41579-020-0394-z) doesn’t mention hipposiderids at all in the text. Anthony et al. 2017 (doi: 10.1093/ve/vex012) highlight Pteropodidae, Miniopteridae, and Vespertilionidae families as being significantly associated with coronavirus detections in Africa and Asia, and Hipposideridae as not significantly associated (and with consistently lower odds ratios for detection than other families) (see Table 2). Drexler et al. 2014 (doi:10.1016/j.antiviral.2013.10.013) list many other bat families in relation to CoVs

(Nycteridae, Pteropodidae, Vespertilionidae, Molossidae, Megadermatidae, Rhinolophidae, and Emballonuridae) and doesn't appear to mention hipposiderids as being any more important. Geldenhuys et al. 2021 (doi:ARTN 93610.3390/v13050936) highlights the overrepresentation of hipposiderids among individuals tested for CoVs (33% of all individual bats tested), yet report higher proportions of detections from other families (Pteropodidae, Molossidae, Miniopteridae, Vespertilionidae, Rhinolophidae, Nycteridae, Rhinonycteridae, and Megadermatidae). Of all bat families with CoVs, hipposiderids have the lowest proportion of detection. L129-131: "By contrast, the MERS-related beta-CoV 2c strain was only diagnosed in the large-eared slit-faced bat *Nycteris macrotis* in Ghana, possibly indicating a narrow host range of the pathogen [Annan et al. 2013 doi:10.3201/eid1903.121503]." The cited paper reports 2c betacoronaviruses in *Nycteris gambiensis*, but does not mention *Nycteris macrotis*. The two species are not synonyms, as far as I am aware. (It also seems a strange conclusion, given that the original citation reports the detection of this CoV clade over 2 different host families, Nycteridae and Vespertilionidae).

Given the clear importance of subadults (and the substantial difference in representation of Hipposideridae vs non-Hipposideridae families), it could be worth presenting the breakdown of adult and subadult captures per species, in Table 1. I'm wondering how many of the non-Hipposideridae individuals were subadult, and whether this might inherently bias results (e.g., if there were no/few subadult individuals from the Emballonuridae, Pteropodidae, Nycteridae, and Rhinolophidae families, then CoV prevalence might be lower with higher diversity metrics purely owing to the lack of subadults). If there are any species for which you didn't capture subadults, are the results consistent if these species are excluded from analyses?

The results as presented are quite difficult to interpret. A suggested alternative structure for any model selection approach: the authors could start section L265-266 with clear statements on which models were ranked as best from the model selection, and present Supplementary Table 5 in the main text so that readers can see how competitive each model was. The authors could include % variation explained per model within this table. Having specified which models were most highly ranked, the authors could then specify the

effect sizes and significance of variables retained in the best models. They could remove Table 2 from the main text (move to the SI), and present most of this information within the text. The effect sizes should be referred to because these are very low for the Shannon diversity metric (the lowest among included variables), even in the most highly ranked models. Then, relating to L286-289 & Supplementary Table 6: The authors should repeat the model selection approach for the Simpson Diversity Index and Species Richness metrics, and present the coefficients for the best ranked models. It is not valid to assume that the AIC ranking will be the same as for the Shannon diversity metric.

Out of curiosity, what do the authors think of the potential effects of co-infection and cross protection dynamics? I wonder whether attempts to analyse the dataset for concepts like the dilution effect, are too much a simplification of bat-virus dynamics where multiple viruses (even beyond those included here) with variable intra- and inter-species transmissions occur.

Minor Comments:

Reviewer 3 asked about how fecal samples were collected, and this was not addressed at all in the rebuttal, nor clarified in the text. How faecal samples were collected could affect interpretations of prevalence (e.g., were bats left to defecate in bags, and if so, for how long might the faeces have been left before being transferred to buffer? Was this consistent across site and sampling events?) This should be clarified in the main text or SI.

The authors report how many bats were captured (e.g., L164 and L222), but how many captured bats had faecal samples taken? It appears from Table 1 that 100% of bats were successfully sampled for faeces, but it would be worth making this clear in the text.

L201-205 & L256-263: testing for correlations between diversity metrics and CoVs seems unnecessary if the diversity metrics are included in the GLMM model selection. I would delete from the manuscript.

L206-211: why have the authors tested CoV as a binomial response? The predictor variables

are calculated per site*sampling period (i.e., are not at an individual level level). The authors could also include CoV prevalence as a response measure.

Table 2 – how did the authors select which models to present in table 2? Presumably they've selected the best model across all susceptible species sets, per CoV. But they've included two models for CoV 2b with non-competitive AIC values: AIC 2149.0 and AIC 2128.0, for full models with H. abae and H. caffer D respectively.

L215-216: What AIC values are considered competitive? (e.g., models within 2 AIC values?) This should be specified.

L215-216: So how many models were included in this candidate set? My impression is quite a number: in the SI 40 models are presented, where only the top 5 are presented for each susceptible species. It would be worth being transparent about this.

L217-218: "...check for multicollinearity using the `check_collinearity()` function in the 'performance' package." The authors need to specify these correlations (presentation could be in the form of a correlation matrix in the SI), what threshold they considered to be highly correlated (e.g., correlation coefficients >0.8), and what they did with highly correlated variables (e.g., exclude one variable from the variable pair). Or else state that no variables were highly correlated. The authors need to be transparent if highly correlated variables were retained in analyses, because this will dramatically effect interpretation of results.

L59 – "Such [a] relationship"

L59 – swap "harbour" for "host". Framing is important. Harbour has negative connotations.

L65-67 & 67– "...significantly influenced..." & "...is influenced..." be more specific in the direction of this relationship.

L81-82: "It is estimated that wild animals are the origin of at least 60% of zoonotic infections in humans." This statement is incorrect. This wording implies that non-wildlife (i.e., domestic animals) are responsible for 40% of zoonotic infections. The reference that is cited

(Jones et al 2008, doi:10.1038/nature06536) states that 60.3% of emerging infectious diseases in humans are zoonotic (i.e., 40% of EIDs are not zoonotic). Of the zoonotic EIDs, 71.8% originate in wildlife (i.e., 28.2% originate in non-wildlife animals).

L99-101: “Although there are roughly 1,400 known species of bats globally, their diversity tends to decrease sharply in anthropogenically modified habitats, often favouring a small number of dominant species.” I don’t know that I agree with the generality of this statement. The effect of anthropogenic modification would depend on context and species. For example, this could be true for e.g., forest or cave dwelling species, where disturbance can drive away sensitive species from forest patches and caves and leave generalists. On the other hand, the presence of anthropogenic structures (e.g., buildings) can promote co-roosting of many species in individual buildings roosts (in contrast to natural roost structures like tree hollows which are often more species specific). I would temper this sentence: “Although there are roughly 1,400 known species of bats globally, their diversity can decrease in anthropogenically modified habitats, and favour a small number of dominant species.”

L111-112: “though viral shedding rates remain high during active infections.” Do you have a citation to support this statement? We often see bats shedding low amounts of virus compared to other animals, which is why bat-borne viruses often need to go through an intermediate amplifying host (e.g., Hendra virus).

L115-116: Not unique to cave-roosting species.

L112-114: What do the authors mean by “particularly high numbers of roundleaf bats” – particularly high in comparison to what? High numbers of roundleaf bats in roosts, compared to other bat species (e.g., *Eidolon helvum*?) High numbers of roundleaf bat species compared to other bat species? High number of roundleaf bat species or abundance in roosts in Ghana, compared to other countries in Africa? Either specify and provide a reference that empirically shows this (the current citation doesn’t appear to provide empirical support for any of these comparisons), or keep more general “including roundleaf bats”.

L139-140: "...and lastly determined the relationship between bat species diversity and CoV prevalence or infection probability..." for statements like this, should clarify that only a subset of CoV clades were tested.

L149-151: "To standardise sampling effort..." there isn't enough information provided in this sentence to ascertain whether the sampling effort was consistent between surveys – e.g., number of nets set? Duration of netting? Number and experience of people netting? (i.e., speed of removal from net) Proportion of entrances blocked by nets? I would remove "To standardise sampling effort..." or give more information. Though, I see at L195 that your approach requires consistent sampling effort across surveys. If this is really important, I would consider elaborating in the SI.

L204: the mention of five missing sampling events comes out of the blue.

L246-248 & L252: "With the exception of one *Macronycteris gigas* and two *Hipposideros abae*, beta-CoV 2bBasal was found almost exclusively in *Hipposideros caffer*. Likewise, beta-CoV 2c occurred almost exclusively in *Nycteris macrotis*...the beta-CoVs 2bBasal and 2c infect single host species." Only 44 *Macronycteris gigas* individuals were captured, compared with 1,167 *Hipposideros caffer*. I don't think this is enough to disregard *Macronycteris gigas* as a potential host species of beta-CoV 2bBasal and beta-CoV 2c. I imagine this statement was made to justify why models were run for CoV 229E-like and CoV 2b only (?) The authors could more simply state that there were fewer detections across species for beta-CoV 2bBasal and beta-CoV 2c, therefore models were run for CoV 229E-like and CoV 2b which had more cross-species detections.

L267: "factors other than", rather than "other factors than"?

L268: "...revealed that the best model..." selection of the best model is in it's self a result, and should be presented before this point.

REVIEWER COMMENTS

Reviewer #2 (Remarks to the Author):

The manuscript submitted by Dr. Meyer and co-authors (manuscript number NCOMMS-23-22117-T) presents the results of a longitudinal study conducted in Ghana over two years. The study focused on surveying bats roosting in caves and examining the presence of alpha- and beta-coronaviruses (CoVs). A total of over 14,000 bat specimens were sampled across 17 sites, but the study focuses on 2362 specimens surveyed on five cave sites. The bat species were identified using external morphology or cytochrome b (cytb) sequencing (for cryptic species), and four CoVs were screened using real-time RT-PCR. CoV prevalence was calculated for each cave at each sampling time point.

To assess the relationship between CoV prevalence and bat species diversity, various diversity and species richness indexes were calculated and tested for correlation. The objective was to evaluate whether the predictions of the dilution effect hypothesis, which suggests higher disease prevalence in less biologically diverse species assemblages, applied to the studied system. Additionally, generalized linear mixed effects models (GLMM) were employed to fit CoV infection probability of the two multi-host pathogens (alpha-CoV 229E-like, and beta-CoV 2b) using the inferred diversity indexes, the relative abundance of the most common and susceptible bat species (*Hipposideros caffer* complex and *H. abae*), and the relative abundance of subadult bats as explanatory variables.

The authors' main conclusions are that there is a connection between bat species diversity and CoV prevalence and that the abundance of the most common bat species and subadults influences the likelihood of CoV infection.

Overall, the manuscript is well-written and well-organized. It presents a detailed account of the study's technical aspects. The analytical approach is appropriate, with a clear presentation of statistical tests, and the conclusions are supported by the results (but see comments below). The manuscript contributes relevant and original findings, as there have been no prior longitudinal studies involving multi-host viruses in multiple independent bat assemblages. This represents an interesting endeavor to analyze the relationship between diversity and disease in bat coronaviruses.

Major comments:

The main point of this study is to test the predictions of the dilution effect hypothesis (DE) in assemblages of cave-dwelling bats. The problem is that the effects of host community composition or the presence of a highly competent species for the pathogen in question are not isolable under the reported results. The authors show that there is a moderate/low negative correlation between CoV prevalence and species diversity. But there is a specific case under study and not all possible cases, and this makes that the decrease in diversity cannot be isolated from the increase in *H. caffer* relative abundance. So, is it diversity, or is it just *H. caffer*, which is the confirmed reservoir for both multi-host CoVs? This has to be problematized since the results reported in this study can be more simply interpreted as in a previous study involving mixed-species roosts in Yunnan province, China, where a greater prevalence of SARSr-CoVs was found when *Rhinolophus sinicus*, a primary host of SARSr-CoVs, was more abundant in the roost than other species (<https://doi.org/10.1371/journal.ppat.1006698>). Thus, the presence of key species (and their density) included in an assemblage may determine pathogen prevalence and not any intrinsic property of biodiversity (<https://doi.org/10.1890/07-1047.1>).

I suggest reorienting the manuscript in terms of the relative abundance of the *H. caffer* complex.

The only aspect that could be at some extent attributable to the DE is the fact that *H. abae*, although being also a potential reservoir for both alpha-CoV 229E-like, and beta-CoV 2b, could function as a less-competent host than *H. caffer*, so when it increases its relative proportion in a given assemblage, the overall prevalence tends to decrease. I suggest the authors discuss this interesting result. But again, it is not about species diversity, but instead of which of these two species is more abundant. I think that the authors should test for other species (individually and grouped as a whole) in the GLMM (and also in correlation analyses) that are not competent hosts. The species *Coleura afra* is likely a candidate of a non-competent host that when it increases its relative abundance, it produces a decrease in CoV prevalence.

However, the authors should discuss the fact that the effects of host community composition or the presence of a highly competent species for the pathogen in question are difficult to isolate (<https://doi.org/10.1017/s0031182012000200>).

As Keesing et al. (2010) point out:

“For example, if those host species most responsible for amplifying the pathogen tend to persist or even thrive as biodiversity is lost, then disease risk will consistently increase as biodiversity declines. On the other hand, if amplifying species tend to disappear as biodiversity declines, then biodiversity loss will tend to reduce disease risk. These hypothetical possibilities indicate the importance of understanding both the non-random sequences by which species are lost from communities, and whether the species that tend to occur only in more species-rich communities tend to amplify or buffer pathogen transmission”.

In this sense, there is no single rule for disease-diversity relationships. The authors claim that more generalist and susceptible hosts are more prone to prevail in the face of biodiversity loss, but this is not the only possible outcome.

In addition, not any modification in the composition of a given community would have the same outcome. The extent to which mechanisms that are affected by species composition relate to biodiversity would depend on how species are added to (or lost from) a community. These assemblies can be substitutive (individuals of a new species replace individuals of existing species), additive (adding new species adds more individuals to a community), or a combination of both. This can only be assessed by estimating absolute species abundances and the authors work with relative abundances. I suggest the authors discuss this aspect in the manuscript.

For the above-mentioned reasons, it is not appropriate to state that this study provides evidence for the dilution effect hypothesis. In the best case, the results are compatible with the DE hypothesis, but it should be remarked that this is purely speculative since there are no counterexamples to prove that there is any intrinsic property of biodiversity operating in the studied system. Therefore, I suggest restricting the interpretation to the abundance of the main reservoir (and to the proportion of subadult specimens) and extending the analysis to non-competent/less-competent species.

We thank the reviewer for the comments. We now communicate the limitations of our observational study, specifically the challenge of disentangling the impacts of changes in species diversity from alterations in species abundance within the community composition. While species diversity provides insight into shifts in the species assemblage, it must be interpreted in the context of changes in the abundance of key species and potentially age-related differences, which may not be adequately captured by diversity indices alone. In the current version of the manuscript, we have made substantial changes to address these concerns and have expanded the discussion based on your valuable suggestions on various aspects.

We have also now conducted additional GLMM analyses involving common but less or non-competent host species, specifically *Coleura afra* and *Nycteris macrotis*, to gain further insights into their potential contributions to the disease dynamics.

After some consideration we decided not to pool species by their genus for subsequent analyses because we lose the information of varying host specificity of different CoVs. Pooling is often done by virus discovery studies, but it actually hinders understanding the mechanisms behind the pattern due to insufficient resolution (Wang et al. 2023 Nat. Commun. <https://www.nature.com/articles/s41467-023-39835-1>). We feel it is very important and a strength of our study not to merge bat species because we have observed differences in host competence to the alpha-CoV229E and beta-CoVs among hipposiderids, reflected in infection numbers and viral shedding rates. These differences are, in part, attributed to immunogenetic variations between the *Hipposideros* spp. (Schmid and Meyer et al. 2023 Mol Ecol, <https://onlinelibrary.wiley.com/doi/epdf/10.1111/mec.16983>). By maintaining resolution at the species level, we can highlight important dynamics, such as the results for *H. abae*, which may not be discernible if we were to group them under a single genus. To justify our choice, we have included a new GLM that clearly shows that among hipposiderids there is variation in competence. Moreover, considering host species allows us to delve deeper into the unique characteristics and responses of individual species. We hope these clarifications demonstrate the scientific rationale behind our decision of sticking to species-level analyses.

Minor comments:

Line 87: remove “this diversity-disease relationship”.

This section was completely rewritten.

Line 90: the DE is not only criticized for its limitations but also its exceptions. The DE is not the only possible outcome of biodiversity loss: no relationship, a context-dependent relationship or amplification effects have been reported in the same manner as DE (see <https://doi.org/10.1371/journal.ppat.1009637>, <https://doi.org/10.1038/s41559-019-1060-6>, and references therein). The authors should present a complete panorama of this topic.

We agree with the reviewers and rewrote the introduction and discussion accordingly. A strict reference limit, however, means some references had to be left out.

Line 189: the reference cited does not apply qRT-PCR, so it is impossible that the authors can take a Ct from this paper. Please be more cautious and responsible in the use of bibliographic references.

Thank you for pointing out this discrepancy. We apologise for the oversight and have double-checked all references. Furthermore, we added a more detailed explanation to the Supplementary Material of how the cut-off Ct value of 38 was determined.

Line 191: the authors should state which is the universe in which the statistical analyses are based (the set of 2362 bats).

Done.

Line 209: I suggest that the authors run the same analysis taking the *H. caffer* complex as a whole, and test if this variable produces stronger correlations and more explanatory models.

We have chosen to maintain resolution at the species level to preserve the unique insights gained through individual species analysis (see response above). Grouping the bat species back to the cryptic complex would indeed obscure valuable information obtained from extensive molecular lab

work on identifying differences in host competence across the complex (Schmid and Meyer et al. 2023 Mol Ecol, <https://onlinelibrary.wiley.com/doi/epdf/10.1111/mec.16983>). This decision is now further supported by additional statistical tests. For more comments on this topic, please refer to the feedback from and reply to Reviewer 4.

Line 259: remove “strong”. If it is moderate, it is not strong.

Amended.

Line 304: a weak/moderate negative correlation.

We added a sentence to that effect.

Lines 334-336: revise taking into consideration the major comments made above. The authors cannot separate the loss in diversity from the increase in the frequency of H. caffer.

We re-evaluated the interpretation of our results accordingly and have made the necessary changes to the relevant paragraphs.

Lines 337-338: the ‘dead end’ explanation is difficult to invoke in pathogens capable of spreading enormous quantities of infective stages into the environment, such as coronaviruses. CoVs are likely to be transmitted passively in co-roosting bats. Long periods of viral shedding in feces, superspreading, and aerosolization, are some of the characteristics that make viral sharing more likely in cave-dwelling bats, independent of their competence (see <https://doi.org/10.3390/d9030035>).

We agree, the phrase has been revised.

Lines 340-343: again, the authors analyze only one possible outcome. It could equally happen that more competent hosts disappear in species-depleted communities, or that the change is additive, substitutive, or a combination of both.

Thanks, we have outlined these other possibilities and complexities associated with pathogen transmission in cave-dwelling bats in the discussion.

Lines 347-348: revise taking into consideration the major comments made above. The authors cannot separate the loss in diversity from the increase in the frequency of H. caffer.

We have carefully revised the relevant sections of the manuscript based on the major comments and concerns you raised.

Line 349: metrics, not metricses.

Corrected.

Lines 349-351: at least for bat CoV 229-E-like it is obvious that metrics that place more weight on host abundance and evenness will have a stronger correlation with CoV prevalence since the only intervening variable found is the abundance of H. caffer. Again, it is not diversity, it is the abundance of H. caffer, the most competent reservoir for these CoVs. The counterexample to prove that it would be a matter of species diversity would be a cave where species diversity is diminished but the dominant species is a less-competent host. In this sense, the authors can

interpret the case of *H. abae*, as a less-competent host, that when increases its density, at the expense of *H. caffer*, induces a decrease in beta-CoV 2b prevalence. But again, this is not a property of biodiversity, but of these two key species in particular.

We have considered and addressed your comments in the revised manuscript.

Line 383: It is not less diverse bat assemblages, but those in which the natural reservoirs (*Hipposideros caffer* complex and *H. abae*) are abundant.

We have adapted the sentence.

Reviewer #4 (Remarks to the Author):

Meyer & Schmid et al. explored the relationship between bat species diversity and coronavirus prevalence / infection likelihood among cave-dwelling bat communities in Ghana. The authors contextualise the study as providing empirical evidence for the “dilution effect” (low disease prevalence in biologically more diverse species assemblages).

I want to commend the authors for the wealth of data they have generated on this system. Longitudinal datasets on bat CoVs are scarce, and the authors present a truly impressive sample set that will generate extremely valuable insights into bat-CoV dynamics. However, Reviewers 2 and 3 have raised several points about the study’s inability to investigate the dilution effect, which I agree with, and which have not been adequately addressed by the authors. I elaborate below, and also give my own separate comments. I hope the authors take the feedback as positive and constructive, as intended.

Thank you for your constructive feedback. We acknowledge the concerns raised by all Reviewers regarding the study's limitations in investigating the dilution effect and addressed these points carefully in our revisions.

General Comments Relating to Previous Reviews:

As highlighted by Reviewers 2 and 3, it is likely that key species abundance – and not assembly diversity – may correlate with CoV prevalence. The authors rebuttals against this comment seem to have missed the point. Reviewer 2 did not state that coronavirus-bat relationships are an unsuitable study system to investigate disease-diversity relationships, only that there needs to be at least one counterexample where prevalence increases as species diversity decreases *and where the most abundant species are different from natural reservoirs* to demonstrate the dilution effect. If this situation is not found, then observed patterns cannot be inferred as the result of changes in biodiversity, because patterns may simply reflect changes to key species abundance.

Consider two hypothetical cave roosts, both with two species each because of high disturbance. In cave one the most abundant species is a generalist and CoV reservoir, in cave two the most abundant species is a generalist but NOT a CoV reservoir. Both caves have the same assembly diversity, but the prevalence of CoVs will be higher in cave one where the abundant species is a CoV reservoir. This hypothetical scenario demonstrates how prevalence patterns can reflect key species abundance, and not changes in assembly diversity as predicted by the dilution effect. This phenomenon is described more succinctly by Reviewer 2: “If those host species most responsible for amplifying the pathogen tend to persist or even thrive as biodiversity is lost, then disease risk will consistently increase as biodiversity declines. On the other hand, if amplifying species tend to disappear as biodiversity declines, then biodiversity loss will tend to reduce disease risk. These

hypothetical possibilities indicate the importance of understanding both the non-random sequences by which species are lost from communities, and whether the species that tend to occur only in more species-rich communities tend to amplify or buffer pathogen transmission". "Thus, the presence of key species (and their density) included in an assemblage may determine pathogen prevalence and not any intrinsic property of biodiversity." (<https://doi.org/10.1890/07-1047.1>).

We are grateful to the reviewer for providing clarification and insight to the interpretation of the previous reviewer 2's responses. It has indeed made our discussion clearer and better contextualized. We have made extensive efforts to broaden the discussion and critically evaluate the assumptions of the dilution effects. We believe our revisions align well with the suggestions from both reviewers.

Looking at figure 1B, the relative abundance of species in the majority of time points, are dominated by *Hipposideridae*. I see only one time point (of 55) where the *Hipposideros caffer* complex is absent (though where *Hipposideros abae* remain). At this time point, there are 5 species present (comparable to other time points), and negligible CoV prevalence of all clades. This doesn't meet the situation as outlined by Reviewer 2, and could actually indicate that the relative abundance of *Hipposideros* spp. drive infection, not assembly diversity.

We acknowledge the limitations of our observational study, particularly the challenge of disentangling the impacts of changes in species diversity from the abundance of competent host species. Based on your comments, we have carefully adapted our interpretations and conclusions to reflect this consideration.

Reviewer 2 highlighted that both the *Hipposideros caffer* complex and *Hipposideros abae* are likely natural reservoirs for the two CoV clades analysed, and suggested that the authors analyse the effect of these hosts as a single set. I agree with this comment, and am surprised that the authors didn't accommodate this seemingly straightforward suggestion. The two CoV clades analysed are predominantly associated with hipposiderid and rhinolophid bats (e.g., Anthony et al. 2017, doi: 10.1093/ve/vex012). This makes it difficult to interpret what the diversity metrics mean in relation to the "dilution effect". E.g., compare two hypothetical groups each with 6 species, but one group is comprised entirely of hipposiderid species, and the other with 1 hipposiderid species and 5 non-hipposiderid species. The two groups should have the same diversity index, but vastly different infection potentials for these CoV clades. The authors have attempted to account for the presence of susceptible species, by including the relative abundance of four key hipposiderid species in GLMMs. However, these species are considered in isolation, so that each model set accounts for the relative abundance of one dominant/susceptible species, but not the others. It is therefore still difficult to interpret what the diversity metrics mean in relation to the "dilution effect". I would recommend that the authors first test to see whether infection of individual bats is predicted by species in this dataset (in particular, test this for the hipposiderid – and maybe rhinolophid – subset, to test for differences between these key species). If infection probability is not different between these species, I would consider them as a single dominant host "group" in the GLMM set of models, for the purpose of testing the "dilution effect", as suggested by Reviewer 2.

Thank you for the suggestion. We have now included a generalized linear model testing whether infection of individual bats is predicted by species identity (and age) on the entire dataset, as well as on the *Hipposideros* subset. This analysis revealed significant differences among the species in terms of infection probability. Additionally, we found differences in the shedding rates as reflected by Ct-values, indicating variations in host competence. We believe these differences are influenced, in

part, by immunogenetic variations in the MHCII complex, as supported by our recently published data (<https://onlinelibrary.wiley.com/doi/epdf/10.1111/mec.16983>). Moreover, as mentioned in our response to Reviewer 2, we believe that grouping the bat species back to the cryptic complex would obscure valuable information gained from extensive molecular lab work that identified differences in host competence across the complex. Maintaining resolution at the species level allows us to preserve the unique insights obtained through individual species analysis and better understand the interactions between CoV strains and hosts.

Furthermore, we want to clarify that the decision to consider each of the species' abundances in separate models was made due to statistical reasons. Due to collinearity among the predictors (please see the two graphs below), we opted to analyse them separately in the GLMMs and corrected for multiple testing, ensuring the robustness and reliability of our findings.

A) Collinearity among variables predicting CoV 229E-like infection likelihood

B) Collinearity among variables predicting CoV 2b infection likelihood

The authors discuss in the rebuttal that the two modelled CoV clades are found in multiple hosts, even outside the Hipposideros complex. I think this is the point (that the CoV clades are potentially associated with multiple species in the dataset, therefore accounting for the relative abundance of any one species in isolation doesn't capture the whole picture). It would be valuable to include information about CoV clade*bat family/species relationships in the text, to better contextualise the specific CoV-bat species relationships included in the manuscript. This is currently missing from the manuscript. I would also encourage the authors to present a tabular breakdown of species captured per session*site in the SI, so that the readers might be able to

interpret species compositions relative to the family groupings. It would also be easier to interpret the species assemblages in Figure 1B if key the susceptible bats (hipposiderid bats) were ordered together at the bottom of bands, and distinctly coloured (e.g., hipposiderid species as shades of red, and non-hipposiderid species as shades of grey).

We would like to point out again that the decision to test species abundance in separate models was driven by statistical considerations, i.e., high VIF factors among predictor variables. As for the CoV clade-bat species interaction, we believe it is not necessary to include it since the specific CoV clade serves as the response variable, and our analysis already accounts for differences in the CoV clade-bat species relationship. Our primary focus was to highlight these differences between CoV clades and *Hipposideros* species, which is effectively addressed in our analysis.

Apart from that, we have taken your suggestion into account and included the tabular breakdown of species captured per season and site. In the Figure 1B, we have ordered the *Hipposideros caffer* complex together following your suggestion, but have chosen not to match colour shades because we want to emphasise the unique differences among the species.

Additionally, in the presented context of disturbance and biodiversity loss, one might interpret hipposiderid bats to be dominant, generalist species, given their high representation in the data. However, it is my impression that African hipposiderid bats are more specialist species that are sensitive to disturbance (e.g., Webala et al. 2004 <https://doi.org/10.1111/j.1365-2028.2004.00505.x>; Wechuli et al 2016 <https://doi.org/10.1111/aje.12376>). This is relevant to the points raised by Reviewers 2 and 3, because one might expect that with disturbance and loss of hipposiderid bats, prevalence of these specific CoV clades should decrease, regardless of biodiversity. If the authors still want to keep the dilution effect narrative, it could be worth clarifying the potential role of hipposiderid bats as a study system for understanding the dilution effect in the context of disturbance and species composition change. It would also be worth presenting some information on the extent of disturbance at sampled sites, within the methods, so that the reader might interpret to what extent this system represents a disturbed habitat.

This is a great suggestion by the reviewer. We have taken it into consideration and included additional information on the extent of disturbance experienced by the bats in their various caves in the Supplementary Information. We now provide the classification of bat caves according to Tanalgo et al. 2018

(<https://www.sciencedirect.com/science/article/abs/pii/S1470160X17307768?via%3Dihub>), as completed by Nkrumah et al. 2021

(<https://journals.sagepub.com/doi/full/10.1177/19400829211034671>). Additionally, we present survey results that highlight the amount and reasons for cave visits by locals, along with a table providing estimated levels of human disturbance from agriculture, pollution, and other factors based on data from Theobald et al. 2020 (<https://essd.copernicus.org/articles/12/1953/2020/>) and 2023 (<https://zenodo.org/record/7534895>).

Regarding the position of *Hipposideros* bats on the generalist-specialist continuum, we recognize that there are likely species-specific differences. For instance, *Hipposideros caffer D* appears to thrive in the most disturbed caves of Kwamang. While this may suggest this species is more amenable to disturbance compared to other species (e.g., *Hipposideros caffer B*), we caution against making generalized statements about generalist/specialist designations, as we lack sufficient ecological information for these species. We have refrained from making general statements about generalist/specialist tendencies in the manuscript and acknowledge that species-specific relationships with CoV prevalence may exist. By maintaining resolution at the species level, we can highlight these unique interactions and avoid covering up valuable insights gained from individual species analyses.

Lastly, I would note that Reviewer 2 referred to the vector-borne pathogen literature by way of illustrating the two mechanisms that can generate the dilution effect (transmission interference and susceptible host regulation). These mechanisms (or certainly the second mechanism) can operate for non-vector disease agents, but the point is that it's context specific and hard to prove empirically. The authors rebut "The concept proposes that the presence of a more diverse array of hosts in a community can reduce the chances of an infected individual encountering a susceptible host, thereby limiting the transmission of disease." This is true, but one must be able to differentiate between the effects of key species abundance and changes in biodiversity, as described above.

We distinguish between mechanisms in the updated manuscript. Our revised manuscript offers a more nuanced discussion, recognizing the complexities of the system.

Overall, I think the bat-CoV dynamics in this dataset are interesting and worthy of publication. I would just reframe the narrative of the paper to remove the focus on testing the "dilution effect" and contextualise the changes as they relate to specific species compositions (host vs non-host). The authors could use this as an opportunity to discuss why the dilution effect is not immediately useful as a generalisable concept (in the sense that disease-diversity patterns must be interpreted with regard to specific host/species assemblages).

We appreciate this comment by the reviewer and have done so in the updated manuscript. Particularly the host specific dynamics at species level are to us very interesting. We discuss the challenges and limitations of the dilution effect as a generalisable concept, emphasising the need for context-specific interpretations of disease-diversity patterns based on unique host/species assemblages and their abundance pattern. Your feedback undoubtedly enhanced the clarity and relevance of our manuscript.

General Comments, New:

I would encourage the authors to check the relevance and accuracy of all citations. There are several citations that are either unrelated to the statement its supporting, or directly counter the statement – e.g, L81-82: "It is estimated that wild animals are the origin of at least 60 % of zoonotic infections in humans" Allen et al. 2017 (doi:10.1038/s41467-017-00923-8) doesn't give a statistic of this nature. It shows the spatial distribution and predictors of zoonoses from wildlife; L112-114: "As the only mammal capable of powered flight, bats are exceptionally mobile and often congregate in roosting caves offering opportunities for intra- and interspecies transmission of pathogens over long distances." Neither reference provided – Letko et al. 2020 (doi:10.1038/s41579-020-0394-z) nor Li et al. 2005 (doi:10.1126/science.1118391) – discuss migration, aggregation, or intra/interspecies transmission. L126-127: "Besides, hipposiderids count amongst the most important reservoirs of CoVs in the paleotropics" Letko et al. 2020 (doi:10.1038/s41579-020-0394-z) doesn't mention hipposiderids at all in the text. Anthony et al. 2017 (doi: 10.1093/ve/vex012) highlight Pteropodidae, Miniopteridae, and Vespertilionidae families as being significantly associated with coronavirus detections in Africa and Asia, and Hipposideridae as not significantly associated (and with consistently lower odds ratios for detection than other families) (see Table 2). Drexler et al. 2014 (doi:10.1016/j.antiviral.2013.10.013) list many other bat families in relation to CoVs (Nycteridae, Pteropodidae, Vespertilionidae, Molossidae, Megadermatidae, Rhinolophidae, and Emballonuridae) and doesn't appear to mention hipposiderids as being any more important. Geldenhuys et al. 2021 (doi:ARTN 93610.3390/v13050936) highlights the overrepresentation of hipposiderids among individuals tested for CoVs (33% of all individual bats tested), yet report

higher proportions of detections from other families (Pteropodidae, Molossidae, Miniopteridae, Vespertilionidae, Rhinolophidae, Nycteridae, Rhinonycteridae, and Megadermatidae). Of all bat families with CoVs, hipposiderids have the lowest proportion of detection.

Thank you for bringing this to our attention. We apologize for any inaccuracies in the citations and their relevance to the statements in the manuscript. We thoroughly reviewed all citations to ensure their accuracy and appropriateness in supporting the corresponding statements.

L129-131: “By contrast, the MERS-related beta-CoV 2c strain was only diagnosed in the large-eared slit-faced bat *Nycteris macrotis* in Ghana, possibly indicating a narrow host range of the pathogen [Annan et al. 2013 doi:10.3201/eid1903.121503].” The cited paper reports 2c betacoronaviruses in *Nycteris gambiensis*, but does not mention *Nycteris macrotis*. The two species are not synonyms, as far as I am aware. (It also seems a strange conclusion, given that the original citation reports the detection of this CoV clade over 2 different host families, Nycteridae and Vespertilionidae).

The reviewer is correct that the Annan et al. paper states that. However, this is actually a mistake in taxonomy assignment in the earlier publication, which Peter Vallo, our co-author and bat taxonomist confirmed. The actual species infected was *Nycteris macrotis*, which has been identified as the species in our caves in Nkrumah et al. 2021

(<https://journals.sagepub.com/doi/full/10.1177/19400829211034671>).

The narrow host range was inferred based on the fact that the virus was detected in *N. macrotis* in Ghana but also in geographically non-overlapping *Pipistrellus* bats from Europe. However, we tempered that statement.

Given the clear importance of subadults (and the substantial difference in representation of Hipposideridae vs non-Hipposideridae families), it could be worth presenting the breakdown of adult and subadult captures per species, in Table 1. I’m wondering how many of the non-Hipposideridae individuals were subadult, and whether this might inherently bias results (e.g., if there were no/few subadult individuals from the Emballonuridae, Pteropodidae, Nycteridae, and Rhinolophidae families, then CoV prevalence might be lower with higher diversity metrics purely owing to the lack of subadults). If there are any species for which you didn’t capture subadults, are the results consistent if these species are excluded from analyses?

We followed the reviewer’s suggestion and constructed a second table based on Table 1, but differentiated by age group. This has been placed in the SI. Apart from *Rousettus aegyptiacus*, which was only caught twice, all species had subadults represented in our dataset.

We observed that subadults generally exhibit lower Ct values (SI Fig 3), and their even distribution among the tested species suggests that this effect is not specific to any particular species.

The results as presented are quite difficult to interpret. A suggested alternative structure for any model selection approach: the authors could start section L265-266 with clear statements on which models were ranked as best from the model selection, and present Supplementary Table 5 in the main text so that readers can see how competitive each model was. The authors could include % variation explained per model within this table. Having specified which models were most highly ranked, the authors could then specify the effect sizes and significance of variables retained in the best models. They could remove Table 2 from the main text (move to the SI), and present most of this information within the text. The effect sizes should be referred to because these are very low for the Shannon diversity metric (the lowest among included variables), even in the most highly ranked models. Then, relating to L286-289 & Supplementary Table 6: The authors should repeat the model selection approach for the Simpson Diversity Index and Species Richness

metrics, and present the coefficients for the best ranked models. It is not valid to assume that the AIC ranking will be the same as for the Shannon diversity metric.

We thank the authors for this suggestion. We agree and have provided additional tables on model ranking, included Species Richness and Simpson Diversity as well as results on all competitive full models as tables in the main text and Supplementary.

Out of curiosity, what do the authors think of the potential effects of co-infection and cross protection dynamics? I wonder whether attempts to analyse the dataset for concepts like the dilution effect, are too much a simplification of bat-virus dynamics where multiple viruses (even beyond those included here) with variable intra- and inter-species transmissions occur.

We appreciate the reviewer's curiosity and agree that bat-virus dynamics can be highly complex, involving multiple viruses with diverse transmission patterns. Co-infection and cross-protection dynamics are undoubtedly important factors that can influence disease outcomes in bat communities. While our study provides insights into the relationships between certain CoV clades and specific bat species, we recognize the need for further research to explore the broader dynamics of co-infection and cross-protection in bat communities. This is also part of an ongoing effort to understand the direct and indirect (e.g. microbiome-mediated) ways in which host genetics determine infection likelihood. At present we do not have a clear answer for your question though. Linked to this, we would like to point out that in previous work co-infection between CoV 229E-like and CoV 2b was less likely, whereas CoV 2b and CoV 2bBasal were found more readily together. This is actually curious because given the evolutionary proximity of the two beta-viruses one would expect cross protection in that case and actually fewer co-infections. We have added a brief statement about this in the text and align it with recent work on co-infections in bats from the Yunnan province (Wang et al. 2023 <https://www.nature.com/articles/s41467-023-39835-1#ref-CR20>; and Hu et al. 2017 <https://journals.plos.org/plospathogens/article?id=10.1371/journal.ppat.1006698>).

Minor Comments:

Reviewer 3 asked about how fecal samples were collected, and this was not addressed at all in the rebuttal, nor clarified in the text. How faecal samples were collected could affect interpretations of prevalence (e.g., were bats left to defecate in bags, and if so, for how long might the faeces have been left before being transferred to buffer? Was this consistent across site and sampling events?) This should be clarified in the main text or SI.

We apologise for missing this question. Like much of the fieldwork information we refer to the study by Nkrumah et al. 2021 (<https://journals.sagepub.com/doi/full/10.1177/19400829211034671>) for such details. Since the reviewer requested it, we now added this information in the SI.

The authors report how many bats were captured (e.g., L164 and L222), but how many captured bats had faecal samples taken? It appears from Table 1 that 100% of bats were successfully sampled for faeces, but it would be worth making this clear in the text.

Yes, of course, the subset we analysed contained only bats successfully sampled for faeces. Accordingly, Table 1 represents information only from bats successfully sampled. From the 14,464 captured bats (Supplementary Table 1), a total of 13,051 faecal samples were collected. We added the information to the SI.

L201-205 & L256-263: testing for correlations between diversity metrics and CoVs seems

unnecessary if the diversity metrics are included in the GLMM model selection. I would delete from the manuscript.

We thank the reviewer for their suggestion. To evaluate the potential relationship between longitudinal changes in species diversity, reflecting species assemblage, and CoV prevalences, we conducted a Spearman rank correlation analysis, which avoids assumptions about the underlying data distribution and the linearity of the relationship between variables (Halliday et. al., 2019). We would like to make sure that the reviewers consider that we aimed to answer different questions with the correlation and the GLMMs. The correlation assesses the relationship between any species diversity mirroring changes in community assemblage and CoV prevalence (being the measure typically employed when testing the diversity-disease relationships), while the GLMMs test the effect of the predictors in individual infection likelihood.

L206-211: why have the authors tested CoV as a binomial response? The predictor variables are calculated per site*sampling period (i.e., are not at an individual level level). The authors could also include CoV prevalence as a response measure.

As per earlier response, we addressed a different question with the GLMMs, namely whether individual infection likelihood, i.e. the probability of an individual having a CoV, differs based on species diversity, abundance of subadults or hipposiderids currently in the cave. The correlations thus address the question whether disease prevalence is linked to bat species assemblage, while the GLMMs are more concerned which properties of the bat species assemblage determine infection likelihood.

Table 2 – how did the authors select which models to present in table 2? Presumably they've selected the best model across all susceptible species sets, per CoV. But they've included two models for CoV 2b with non-competitive AIC values: AIC 2149.0 and AIC 2128.0, for full models with H. abae and H. caffer D respectively.

In Table 2, we chose to present the summaries of the full models in the main text based on the significance of predictors of species abundances. These full models provided valuable insights into the relationships between species abundances and coronavirus prevalence. To provide more comprehensive information, we now listed the results of all other models and variables in the SI.

L215-216: What AIC values are considered competitive? (e.g., models within 2 AIC values?) This should be specified.

In our analysis, we considered all models with a ΔAIC_c less than or equal to 2 as competitive. For comparability, we opted for the complete model in cases where the complete model was maybe not considered the best but still within the ΔAIC_c of 2. This information has been added to the text to provide clarity on how we determined the competitive models. To further enhance transparency and facilitate a comprehensive understanding of our model selection process, we have included a table in the SI.

L215-216: So how many models were included in this candidate set? My impression is quite a number: in the SI 40 models are presented, where only the top 5 are presented for each susceptible species. It would be worth being transparent about this.

In total, we ran 36 models during our analysis and we have incorporated this information in the revised manuscript. This included six models per diversity index due to only including one bat species at a time, and we examined three diversity indices for each of the two CoV strains.

L217-218: “...check for multicollinearity using the check_collinearity() function in the ‘performance’ package.” The authors need to specify these correlations (presentation could be in the form of a correlation matrix in the SI), what threshold they considered to be highly correlated (e.g., correlation coefficients >0.8), and what they did with highly correlated variables (e.g., exclude one variable from the variable pair). Or else state that no variables were highly correlated. The authors need to be transparent if highly correlated variables were retained in analyses, because this will dramatically effect interpretation of results.

Indeed, we used the 'check_collinearity()' function from the 'performance' package to assess multicollinearity among predictors. We found moderate (VIF factor 5-10) to high (VIF factor ≥ 10) levels of correlation among some variables why we made the decision to run separate models, with each diversity index and species abundance included individually in its own model. We would like to emphasise that all variables in all 36 models exhibited low correlation (VIF below 5). This information has been added to the manuscript to enhance transparency and demonstrate that multicollinearity concerns were appropriately addressed.

L59 – “Such [a] relationship”

Thanks.

L59 – swap “harbour” for “host”. Framing is important. Harbour has negative connotations.

We see the reviewers point. Nevertheless, consulting with a native, we thought “harbour” more accurately describes the temporary nature of viral infections, whereas host we associate with a persistent infection. Its likely debatable which better matches viral infections in bats. But certainly, either has a negative connotation.

L65-67 & 67– “...significantly influenced...” & “...is influenced...” be more specific in the direction of this relationship.

We have updated the sentence to provide more explicit descriptions.

L81-82: “It is estimated that wild animals are the origin of at least 60% of zoonotic infections in humans.” This statement is incorrect. This wording implies that non-wildlife (i.e., domestic animals) are responsible for 40% of zoonotic infections. The reference that is cited (Jones et al 2008, doi:10.1038/nature06536) states that 60.3% of emerging infectious diseases in humans are zoonotic (i.e., 40% of EIDs are not zoonotic). Of the zoonotic EIDs, 71.8% originate in wildlife (i.e., 28.2% originate in non-wildlife animals).

We altered the statement accordingly. Thanks for pointing this out.

L99-101: “Although there are roughly 1,400 known species of bats globally, their diversity tends to decrease sharply in anthropogenically modified habitats, often favouring a small number of dominant species.” I don’t know that I agree with the generality of this statement. The effect of anthropogenic modification would depend on context and species. For example, this could be true for e.g., forest or cave dwelling species, where disturbance can drive away sensitive species from forest patches and caves and leave generalists. On the other hand, the presence of anthropogenic structures (e.g., buildings) can promote co-roosting of many species in individual buildings roosts (in contrast to natural roost structures like tree hollows which are often more species specific). I would temper this sentence: “Although there are roughly 1,400 known species of bats globally,

their diversity can decrease in anthropogenically modified habitats, and favour a small number of dominant species.”

We thank the reviewer for their suggestion and amended the sentence. Instead of can we placed “might” though.

L111-112: “though viral shedding rates remain high during active infections.” Do you have a citation to support this statement? We often see bats shedding low amounts of virus compared to other animals, which is why bat-borne viruses often need to go through an intermediate amplifying host (e.g., Hendra virus).

This sentence was not retained in the altered text.

L115-116: Not unique to cave-roosting species.

We changed this to roosting community since we agree with the reviewer that this is not a cave specific phenomenon.

L112-114: What do the authors mean by “particularly high numbers of roundleaf bats” – particularly high in comparison to what? High numbers of roundleaf bats in roosts, compared to other bat species (e.g., Eidolon helvum?) High numbers of roundleaf bat species compared to other bat species? High number of roundleaf bat species or abundance in roosts in Ghana, compared to other countries in Africa? Either specify and provide a reference that empirically shows this (the current citation doesn’t appear to provide empirical support for any of these comparisons), or keep more general “including roundleaf bats”.

Thanks for the suggestions. We followed suit.

L139-140: “...and lastly determined the relationship between bat species diversity and CoV prevalence or infection probability...” for statements like this, should clarify that only a subset of CoV clades were tested.

We amended the sentence.

L149-151: “To standardise sampling effort...” there isn’t enough information provided in this sentence to ascertain whether the sampling effort was consistent between surveys – e.g., number of nets set? Duration of netting? Number and experience of people netting? (i.e., speed of removal from net) Proportion of entrances blocked by nets? I would remove “To standardise sampling effort...” or give more information. Though, I see at L195 that your approach requires consistent sampling effort across surveys. If this is really important, I would consider elaborating in the SI.

We removed the part of the sentence as per your suggestion and added more information to the SI. As per SI more information on the sampling procedure was specified in Nkrumah et al. 2021 (<https://journals.sagepub.com/doi/full/10.1177/19400829211034671>).

L204: the mention of five missing sampling events comes out of the blue.

We now mention them when we describe the sample collection.

L246-248 & L252: “With the exception of one *Macronycteris gigas* and two *Hipposideros abae*,

beta-CoV 2bBasal was found almost exclusively in *Hipposideros caffer*. Likewise, beta-CoV 2c occurred almost exclusively in *Nycteris macrotis*...the beta-CoVs 2bBasal and 2c infect single host species." Only 44 *Macronycteris gigas* individuals were captured, compared with 1,167 *Hipposideros caffer*. I don't think this is enough to disregard *Macronycteris gigas* as a potential host species of beta-CoV 2bBasal and beta-CoV 2c. I imagine this statement was made to justify why models were run for CoV 229E-like and CoV 2b only (?) The authors could more simply state that there were fewer detections across species for beta-CoV 2bBasal and beta-CoV 2c, therefore models were run for CoV 229E-like and CoV 2b which had more cross-species detections.

We followed your suggestion and altered the text accordingly. A few lines down we also refrained from coining beta-CoV 2bBasal and 2c a single host pathogen.

L267: "factors other than", rather than "other factors than"?

Thanks. The sentence had changed in the revised manuscript.

L268: "...revealed that the best model..." selection of the best model is in it's self a result, and should be presented before this point.

We agree that the model selection process is important and provided a table to that end. We have extended to all competitive full models run with the dredge function. However, we don't think the precise details of each model is of interest to the wider readership, which is why we placed those results in the SI.

We are very grateful to the reviewers for their thoughtful comments and suggestions!

REVIEWER COMMENTS

Reviewer #2 (Remarks to the Author):

The authors have effectively addressed most of the concerns raised in the previous manuscript revision. However, some issues still persist due to the arbitrary emphasis on species biodiversity rather than the abundance of key host species as explanatory factors for CoV prevalence.

I recommend removing the reference to the correlation between bat diversity and CoV prevalence from the abstract and summary, and approaching it cautiously in the discussion section, as the correlation appears to be moderate to weak. In alignment with the consensus among all reviewers and with the explicit agreement of the authors, it is advisable to shift the focus away from species diversity and emphasize the importance of key species abundance.

The authors have expressed their support for this perspective in their rebuttal letter, stating:

“It must be interpreted in the context of changes in the abundance of key species and potentially age-related differences, which may not be adequately captured by diversity indices alone”

“We acknowledge the limitations of our observational study, particularly the challenge of disentangling the impacts of changes in species diversity from the abundance of competent host species”

In relation to this part of the rebuttal letter:

“After some consideration we decided not to pool species by their genus for subsequent analyses because we lose the information of varying host specificity of different CoVs. Pooling is often done by virus discovery studies, but it actually hinders understanding the

mechanisms behind the pattern due to insufficient resolution (Wang et al. 2023 Nat. Commun.

<https://www.nature.com/articles/s41467-023-39835-1>). We feel it is very important and a strength of our study not to merge bat species because we have observed differences in host competence to the alpha-CoV229E and beta-CoVs among hipposiderids, reflected in infection numbers and viral shedding rates. These differences are, in part, attributed to immunogenetic variations between the *Hipposideros* spp. (Schmid and Meyer et al. 2023 Mol Ecol,

<https://onlinelibrary.wiley.com/doi/epdf/10.1111/mec.16983>). By maintaining resolution at the

species level, we can highlight important dynamics, such as the results for *H. abae*, which may not be discernible if we were to group them under a single genus. To justify our choice, we have included a new GLM that clearly shows that among hipposiderids there is variation in competence. Moreover, considering host species allows us to delve deeper into the unique characteristics and responses of individual species. We hope these clarifications demonstrate the scientific rationale behind our decision of sticking to species-level analyses”.

What I recommended was not to group all *Hipposideros* species at the genus level but rather to combine the cryptic lineages within the *H. caffer* complex. The fact that these lineages can be differentiated through mtDNA genotyping does not preclude the possibility of ongoing gene flow between them. These lineages might still belong to a single species or represent hybridizing sister species, and this cannot be definitively ruled out by relying solely on mtDNA markers. Given their close relationship and their competence as hosts for the two multi-host viruses under study, they can be analyzed as a unified entity. In this context, *H. abae*, which is less closely related and less competent as a host, would remain a separate entity in the analysis.

Minor comments

Discussion

Lines 313-316:

The authors should consistently underscore the modest/limited strength of the correlation identified between bat species diversity and CoV prevalence in all references to this relationship.

Lines 395-401:

I suggest concluding the presented concept by emphasizing that CoV prevalence within the studied system is contingent upon the abundance of key species rather than being driven by any inherent characteristic of biodiversity.

Summary:

Once more, it would be advantageous for the authors to focus on the genuine connection, which is not the feeble correlation between bat species diversity and CoV prevalence, but rather the correlation between key species abundance, alongside with the presence of subadults, and CoV prevalence.

Reviewer #4 (Remarks to the Author):

Meyer & Schmid et al. explored the relationship between bat species diversity and coronavirus prevalence / infection likelihood among cave-dwelling bat communities in Ghana. The authors have made major changes to the manuscript to incorporate feedback from reviewers, and I commend them in adopting feedback re: the dilution effect, and in crafting an interesting narrative on viral infections by specific species compositions. I want to again reiterate that this is an impressive sample set that will generate extremely valuable insights into bat-CoV dynamics. I have two remaining “major comments” and several “minor comments” that I detail below.

Major Comments:

I appreciate that the authors have expanded their explanation of model selection, but I

think this needs to be corrected, and specified in more detail at L214-216. It was only through the response to reviewer comments that I could understand how the models presented at L285-306 were selected, and this seems to be a selection of convenience, rather than a statistically supported selection. For instance, at L287-288 the authors state that “model selection revealed that the best model [for alpha-CoV 229E-like] explained approximately 62% of the variation in infection probability” yet I see in Supplementary Table 12 that there was no single best model for alpha-CoV 229E-like, but 18 equally competitive models. The same is true for beta-CoV 2b. In the response to reviewer comments, the author’s explain that the selection of the best model was “based on the significance of predictors of species abundances”, but this isn’t how model selection should work. Instead of picking just one model from this competitive set, the authors should use multimodel inference and model averaging to estimate the relative importance of variables, taking into account information from all competitive models. The author’s could refer to Symonds and Moussalli (2011) ‘A brief guide to model selection, multimodel inference and model averaging in behavioural ecology using Akaike’s information criterion’ for more information (DOI 10.1007/s00265-010-1037-6). When reporting results, the authors need to state that there were 18 competitive models and that the variation explained by these models ranged from X-X%. They can then describe the interpretation of specific coefficients from model averaging.

I would urge the authors to carefully consider the framing of bat-associated diseases throughout the manuscript, especially in key areas like the abstract. Sentences like those at L61 (“Bats harbour, disperse and transmit many pathogens, including several with zoonotic potential...”) and L104 (“...plethora of disease agents...”) poorly contextualize the realized risk of bats, and can propagate unwarranted negative attitudes, and lead to direct persecution and erosion of local support for bat conservation efforts. The importance of message framing is being increasingly recognised and pushed by researchers in this space, especially in the wake of COVID-19. I would recommend that the authors include statements explaining that if most bat species are left alone, they present little, if any, risk to human health. I would recommend the authors avoid words and phrases with active connotations when describing transmission from bats (to avoid the interpretation that bats actively seek to host and spread pathogens). For example, L102-103 “[Several hypotheses

exist as to why many] pathogens with zoonotic potential originate in and are spread by bats.” could be better phrased as “[Several hypotheses exist as to why many] pathogens are detected in bats.”. Similarly, “harbour” has a more active connotation than “host” and should be avoided. The messaging of the paper could also be balanced by emphasizing the direct, and indirect, health benefits that bats provide to human populations. The messaging around habitat/biodiversity loss and association with disease is good in this respect, but some areas could be improved as described above. The authors could refer to the 2020 publication by MacFarlane and Rocha, ‘Guidelines for communicating about bats to prevent persecution in the time of COVID-19’ for more information (doi: 10.1016/j.biocon.2020.108650).

Minor Comments:

I would suggest removing sensationalist adjectives – e.g., L97 “enormous”, L104 “plethora”,

L96: “...in [the] case of...”

L101-103: These hypotheses explain why bats host a high number of pathogens, period, not specifically pathogens with zoonotic potential. That said, the hypotheses that follow (L103-110) are unbalanced – research also suggests there is NO clear indication of whether bats are “special” for zoonotic viruses. The authors should also recognize these arguments – bats are highly speciose in comparison to other taxonomic groups, so the high number of pathogens detected may be proportional to their species richness. In addition, other taxonomic groups have had very little viral discovery research conducted on them, and with more dedicated research these groups may also turn out to be important for zoonoses. The authors could refer to Olival et al. 2015, ‘Are Bats Really “Special” as Viral Reservoirs? What We Know and Need to Know’ (<https://doi.org/10.1002/9781118818824.ch11>).

L149-150: More information needed on the metal rings – size, type, brand?

L147-148: There is little consensus on the definition and markers of juvenile vs subadult bats, yet these stages are likely very different in their susceptibility to disease. The authors

should be more specific here on how they determined these to be subadults and not juveniles (e.g., define that a subadult is a volant non-adult). I wonder whether they would be better to refer to non-adults as simply “immature”, though, to avoid confusion when comparing across studies.

L151: More information needed on the ethanol – what % ethanol?

L182: More information needed on the RNA copies/ul – how was this viral load quantified (e.g., droplet digital PCR?) and was this quantified in the same laboratory, and therefore directly comparable to the Ct value, or taken from another paper? Ct values alone are not directly comparable across laboratories. This could be specified at first mention, e.g., “...equivalent to >15 CoV-182 RNA copies/ μ L in our assays.” This information is not currently provided in the Supplementary Material.

L200: The word “rate” has a specific meaning (is a quotient of two quantities in different units), and I would caution the authors to be intentional in its use. “Shedding rate” here implies shedding over time, but the data in question are singular Ct values. I suggest replacing all mention of “shedding rate” with “viral load” (but be sure to establish that Ct is a proxy for viral load).

L208-213: I understand now that the response variable is at the individual level, while the explanatory variables are at the sampling level (i.e., unique site/time). However, given there are no explanatory variables at the individual level, why don't the authors use CoV infection prevalence per site/time combination, instead of CoV infection probability per individual? My concern is that the sample size is artificially inflated – for example, if there were 500 bats from a single site/time combination, the model will fit the exact same values for explanatory variables 500 times. Unless I have misunderstood the structure of the model?

L236-264: “...and subadults...” – is this ALL subadults, or subadults of the named species?

L295-296: “...the best models incorporating Shannon diversity explained... and retained Shannon diversity...” – were these models presented specifically because they retained

Shannon diversity? If so, the fact that they retained Shannon diversity isn't a result (i.e., doesn't need to be repeated the second time).

L310: remove "the" in front of "species communities"

L329-361: I found the shift to "chiropterans" from just "bats" slightly odd.

346-357: Confusing structure with "though" and "while" in the same sentence, and with the placement of the citation. Also "co-occur" and "together" mean the same thing, so one of these words is redundant. Suggest: "We had previously found only beta-CoVs 2b and 2bBasal to co-occur frequently [citation], but here found that co-infections of beta-CoV 2b and alpha-CoV 229E-like are less likely."

L349-350: "...More than 40% of locals..." this study didn't survey ALL locals. Its 40% of survey respondents, which could be a biased representation of locals. Suggest "...40% of local survey respondents..."

L352-353: Not all bat species

L354-361: The authors haven't analysed the association with disturbance in this manuscript, and looking at the supplementary information, it seems that the disturbance to the sampled caves was comparable (i.e., there was no pristine, or minimally impacted cave to compare with). Be careful not to imply these relationships in this section.

L363 & throughout – be careful using the word "disease" when referring to bat pathogens. The pathogens in this paper don't cause disease to bats

389: host[s]

391: instance, not instances

L407: Similar to a previous comment, I would consider referring to "immature individuals"

instead of “juveniles”

Supplementary Table 2 isn't interpretable without referring to the cited manuscripts. It would be worth elaborating on how categories were assigned, especially to explain Biotic Vulnerability Scores vs Biotic Vulnerability Index (e.g., the key on L66-70 states that a score between 1-1.99 = index A, yet the table shows that caves with scores of 1.6 and 1.7 have an index of B), and to explain the Bat Cave Vulnerability Index.

Supplementary Table 7: the number of individuals and subadults should be presented without a decimal place

Supplementary Table 12: include the proportion of variation explained by the model in this table.

REVIEWER COMMENTS

Reviewer #2 (Remarks to the Author):

The authors have effectively addressed most of the concerns raised in the previous manuscript revision. However, some issues still persist due to the arbitrary emphasis on species biodiversity rather than the abundance of key host species as explanatory factors for CoV prevalence.

I recommend removing the reference to the correlation between bat diversity and CoV prevalence from the abstract and summary, and approaching it cautiously in the discussion section, as the correlation appears to be moderate to weak. In alignment with the consensus among all reviewers and with the explicit agreement of the authors, it is advisable to shift the focus away from species diversity and emphasize the importance of key species abundance.

The authors have expressed their support for this perspective in their rebuttal letter, stating:

“It must be interpreted in the context of changes in the abundance of key species and potentially age-related differences, which may not be adequately captured by diversity indices alone”

“We acknowledge the limitations of our observational study, particularly the challenge of disentangling the impacts of changes in species diversity from the abundance of competent host species”

We express our gratitude to the reviewer for providing another round of valuable feedback and insightful comments to enhance the quality of our manuscript.

We have conscientiously tackled the concerns raised during the previous manuscript revision, as the reviewer acknowledged himself, resulting in significant revisions to refocus the manuscript towards the role of species abundances over the role of diversity in general. As the reviewer has recognised, we moved away from emphasising species diversity as explanatory factor for CoV. Already in the previous version we have eliminated the reference to the correlation between species diversity and CoV prevalence in the abstract, and specifically emphasise that the difference in CoV is linked to changes in the relative abundance of competent hosts: “Broadly, bat species varied in CoV competence, and highly competent species were dominant in less diverse communities, leading to increased CoV prevalence in less diverse bat assemblages.” Additionally, we also removed any reference to the correlation from the summary, which now reads as: “In summary, shifts in bat community assemblages likely determine CoV prevalence in disturbed cave sites in central Ghana. We emphasise that the abundance of competent bat species and subadults are key ecological drivers of CoV infection likelihood.” This correlation is no longer featured in the abstract and summary sections of the manuscript.

In relation to this part of the rebuttal letter:

“After some consideration we decided not to pool species by their genus for subsequent analyses because we lose the information of varying host specificity of different CoVs. Pooling is often done by virus discovery studies, but it actually hinders understanding the mechanisms behind the pattern due to insufficient resolution (Wang et al. 2023 Nat. Commun.

<https://www.nature.com/articles/s41467-023-39835-1>). We feel it is very important and a strength of our study not to merge bat species because we have observed differences in host competence to the alpha-CoV229E and beta-CoVs among hipposiderids, reflected in infection numbers and viral shedding rates. These differences are, in part, attributed to immunogenetic variations between the *Hipposideros* spp. (Schmid and Meyer et al. 2023 Mol Ecol,

<https://onlinelibrary.wiley.com/doi/epdf/10.1111/mec.16983>). By maintaining resolution at the species level, we can highlight important dynamics, such as the results for *H. abae*, which may not be discernible if we were to group them under a single genus. To justify our choice, we have included a new GLM that clearly shows that among hipposiderids there is variation in competence. Moreover, considering host species allows us to delve deeper into the unique characteristics and responses of individual species. We hope these clarifications demonstrate the scientific rationale behind our decision of sticking to species-level analyses”.

What I recommended was not to group all *Hipposideros* species at the genus level but rather to combine the cryptic lineages within the *H. caffer* complex. The fact that these lineages can be differentiated through mtDNA genotyping does not preclude the possibility of ongoing gene flow between them. These lineages might still belong to a single species or represent hybridizing sister species, and this cannot be definitively ruled out by relying solely on mtDNA markers. Given their close relationship and their competence as hosts for the two multi-host viruses under study, they can be analyzed as a unified entity. In this context, *H. abae*, which is less closely related and less competent as a host, would remain a separate entity in the analysis.

We thank the reviewer for this clarification but disagree strongly with the suggestion of grouping species and considering the cryptic lineages within the *Hipposideros caffer* complex as a unified entity. We provide a list of the various lines of evidence that support our decision to maintain taxonomic resolution at the species level.

- In the work “Concordant patterns of genetic, acoustic, and morphological divergence in the West African Old World leaf-nosed bats of the *Hipposideros caffer* complex” conducted by Baldwin et al. in 2021 (<https://onlinelibrary.wiley.com/doi/full/10.1111/jzs.12506>), the presence of four established mtDNA lineages within the *Hipposideros caffer* complex was confirmed. This genetic divergence was further substantiated through the analysis of nuclear microsatellite data, differences in frequencies of echolocation calls, and morphometric measures. This suggests a genetic, ecological and morphological distinction between the species.
- Additionally, our research extended into investigating immunogenetic differences across these species by examining the MHC class II genes. Our recent publication, “MHC class II genes mediate susceptibility and resistance to coronavirus infections in bats” published in May of this year in *Molecular Ecology* (<https://onlinelibrary.wiley.com/doi/full/10.1111/mec.16983>), provided evidence of significant differences in MHCII allelic and functional diversity among these species. This suggests these bats are immunogenetically distinct.
- Furthermore, in our current manuscript, we present not only variations in CoV infection numbers but also differences in viral load by means of Ct values, both of which are crucial indicators of host competence. This suggests the species vary in pathogen resistance.

Collectively, these lines of evidence strongly support the assertion that maintaining a high level of taxonomic resolution is essential for our study. Comprehending the fundamental phylogenetic relationships among bats and achieving species-level identification is crucial for drawing co-evolutionary insights into virus transmission across diverse bat families and species. We believe that achieving such resolution is not only important for our research but also a broader research aim that should be pursued in disease ecology. Given that biodiversity levels are often underestimated, particularly in tropical ecosystems and in species groups with a high level of cryptic diversity (e.g., Hipposiderids; Foley et al. 2017 *Acta Chiropterologica*; <https://bioone.org/journals/acta-chiropterologica/volume-19/issue-1/15081109ACC2017.19.1.001/Towards-Navigating-the-Minotaurs-Labyrinth--Cryptic-Diversity-and-Taxonomic/10.3161/15081109ACC2017.19.1.001.short>), we consider it essential to maintain a high taxonomic resolution to fully understand and appreciate the unique characteristics and responses of individual species in the context of disease ecology.

Minor comments

Discussion

Lines 313-316:

The authors should consistently underscore the modest/limited strength of the correlation identified between bat species diversity and CoV prevalence in all references to this relationship.

Lines 395-401:

I suggest concluding the presented concept by emphasizing that CoV prevalence within the studied system is contingent upon the abundance of key species rather than being driven by any inherent characteristic of biodiversity.

Summary:

Once more, it would be advantageous for the authors to focus on the genuine connection, which is not the feeble correlation between bat species diversity and CoV prevalence, but rather the correlation between key species abundance, alongside with the presence of subadults, and CoV prevalence.

We have made revisions in accordance with the suggestions to consistently highlight the modest or limited strength of the correlation between bat species diversity and CoV prevalence throughout the manuscript. Furthermore, we have incorporated an additional statement regarding the challenge of disentangling host abundance from inherent biodiversity characteristics in our study system (L391). Thank you once again for the valuable input, which has improved the clarity and precision of our manuscript.

Reviewer #4 (Remarks to the Author):

Meyer & Schmid et al. explored the relationship between bat species diversity and coronavirus prevalence / infection likelihood among cave-dwelling bat communities in Ghana. The authors have made major changes to the manuscript to incorporate feedback from reviewers, and I commend them in adopting feedback re: the dilution effect, and in crafting an interesting narrative on viral infections by specific species compositions. I want to again reiterate that this is an impressive sample set that will generate extremely valuable insights into bat-CoV dynamics. I have two remaining “major comments” and several “minor comments” that I detail below.

We genuinely appreciate the comprehensive and insightful feedback that has been shared regarding our manuscript. We highly value the reviewer's input and recognition of our commitment to addressing previous comments and enhancing the quality of our study. We are sincerely grateful for any additional valuable feedback that contributes to further improvements of our research.

Major Comments:

I appreciate that the authors have expanded their explanation of model selection, but I think this needs to be corrected, and specified in more detail at L214-216. It was only through the response to reviewer comments that I could understand how the models presented at L285-306 were selected, and this seems to be a selection of convenience, rather than a statistically supported selection. For instance, at L287-288 the authors state that “model selection revealed that the best model [for alpha-CoV 229E-like] explained approximately 62% of the variation in infection probability” yet I see

in Supplementary Table 12 that there was no single best model for alpha-CoV 229E-like, but 18 equally competitive models. The same is true for beta-CoV 2b. In the response to reviewer comments, the author's explain that the selection of the best model was "based on the significance of predictors of species abundances", but this isn't how model selection should work. Instead of picking just one model from this competitive set, the authors should use multimodel inference and model averaging to estimate the relative importance of variables, taking into account information from all competitive models. The author's could refer to Symonds and Moussalli (2011) 'A brief guide to model selection, multimodel inference and model averaging in behavioural ecology using Akaike's information criterion' for more information (DOI 10.1007/s00265-010-1037-6). When reporting results, the authors need to state that there were 18 competitive models and that the variation explained by these models ranged from X-X%. They can then describe the interpretation of specific coefficients from model averaging.

Thank you for your comment, and we appreciate the attention to detail in our model selection process. It's important to emphasise that our model selection was not a matter of convenience but was based on rigorous scientific and statistical criteria. We acknowledge that our presentation in the main text may have led to confusion, and our primary objective is to provide a clear and comprehensive clarification of our approach. In addition to our manuscript revisions aimed at improving clarity (L214-225, L290-312 and Supplementary Table 12), we have also detailed our approach in this rebuttal letter to further elucidate our methodology.

Due to the collinearity issue among the explanatory variables, we adopted a specific strategy for constructing our models. Our primary focus was on addressing a biological question related to the potential significance of certain predictors in infection likelihood. To this end, we structured all our models around the inclusion of the predictors „species diversity“, “relative abundance of subadults” and, “species abundance”. To avoid collinearity among the predictors, only one of the different diversity indices (Shannon, Simpson, and Species Richness), the relative abundance of subadults, and the relative abundance of one species was included. This approach served as the foundation for all our models.

We then applied the dredge function to models that contained 1) a diversity measure (but never multiple at the same time), 2) the relative abundance of a specific bat species (but never multiple at the same time) and the relative abundance of subadults. The dredge results were then assessed and we found that all full models were competitive with ΔAIC_c values consistently falling within the threshold of $\Delta AIC_c \leq 2$. Our interpretation and reporting of the results are based on the performance of these individual models, each contributing to an understanding of the observed variation.

Nevertheless, we conducted model averaging for all competitive models listed in Supplementary Table 12 and confirmed that this did not alter the results. To facilitate easy reference and accessibility, we have incorporated the model averaging into our R script, which is available on GitHub. Our intention in presenting these full models in Supplementary Table 12 was not to imply that we followed a traditional model selection process aimed at identifying a single best model among them. Instead, our primary goal was to address a specific biological question by comparing different models with the same structure.

In brief, the full model, initially designed to address a specific biological query, was always competitive when using AIC criteria, and our results and interpretation remained unchanged even if model averaging was applied. This encouraged us that our models are a sound representation of both biologically meaningful and statistically reasonable considerations when examining the impact of the relative abundance of the respective species in relation to species diversity and the relative abundance of subadults.

We genuinely appreciate the reviewer's diligence and recognize the complexities and potential challenges surrounding this topic in ecology. Their feedback has motivated us to provide a clearer explanation, enhanced transparency, and improved visualization. We hope this clarification addresses any concerns regarding our model selection process and its interpretation.

I would urge the authors to carefully consider the framing of bat-associated diseases throughout the manuscript, especially in key areas like the abstract. Sentences like those at L61 (“Bats harbour, disperse and transmit many pathogens, including several with zoonotic potential...”) and L104 (“...plethora of disease agents...”) poorly contextualize the realized risk of bats, and can propagate unwarranted negative attitudes, and lead to direct persecution and erosion of local support for bat conservation efforts. The importance of message framing is being increasingly recognised and pushed by researchers in this space, especially in the wake of COVID-19. I would recommend that the authors include statements explaining that if most bat species are left alone, they present little, if any, risk to human health. I would recommend the authors avoid words and phrases with active connotations when describing transmission from bats (to avoid the interpretation that bats actively seek to host and spread pathogens). For example, L102-103 “[Several hypotheses exist as to why many] pathogens with zoonotic potential originate in and are spread by bats.” could be better phrased as “[Several hypotheses exist as to why many] pathogens are detected in bats.”. Similarly, “harbour” has a more active connotation than “host” and should be avoided. The messaging of the paper could also be balanced by emphasizing the direct, and indirect, health benefits that bats provide to human populations. The messaging around habitat/biodiversity loss and association with disease is good in this respect, but some areas could be improved as described above. The authors could refer to the 2020 publication by MacFarlane and Rocha, ‘Guidelines for communicating about bats to prevent persecution in the time of COVID-19’ for more information (doi: 10.1016/j.biocon.2020.108650).

We agree wholeheartedly with your concerns about the framing of bat-associated diseases in our manuscript. The last thing we intend is to create a negative attitude towards bats. We are fully committed to addressing this issue and promoting proper communication in this context. To address these concerns, we diligently worked on the manuscript to ensure that it strikes a balanced tone (e.g. L60-61, L97-99, L102-107, L436-443). We placed more emphasis on the crucial ecological role that bats play and incorporated additional statements to highlight this aspect. It is of utmost importance to us that the primary conclusion of our paper is the conservation of bats. Thank you for bringing this to our attention, and we are more than willing to make the necessary adjustments to convey a more positive and conservation-oriented message.

Minor Comments:

I would suggest removing sensationalist adjectives – e.g., L97 “enormous”, L104 “plethora”,

We have removed these adjectives to ensure a more balanced and objective tone in our manuscript.

L96: “...in [the] case of...”

Thanks, we have added the missing article.

L101-103: These hypotheses explain why bats host a high number of pathogens, period, not specifically pathogens with zoonotic potential. That said, the hypotheses that follow (L103-110) are unbalanced – research also suggests there is NO clear indication of whether bats are “special” for zoonotic viruses. The authors should also recognize these arguments – bats are highly speciose in comparison to other taxonomic groups, so the high number of pathogens detected may be proportional to their species richness. In addition, other taxonomic groups have had very little viral discovery research conducted on them, and with more dedicated research these groups may also turn out to be important for zoonoses. The authors could refer to Olival et al. 2015, ‘Are Bats Really

“Special” as Viral Reservoirs? What We Know and Need to Know’
(<https://doi.org/10.1002/9781118818824.ch11>).

We have taken these suggestions into account and have worked on revising the manuscript to provide a more balanced perspective (L102-107). As mentioned in our previous response, we aimed to emphasise the important ecological role of bats and the urgent need for their conservation. Thank you also for highlighting the need to recognise the factors related to species richness and research bias - we have incorporated these considerations into the revised manuscript.

L149-150: More information needed on the metal rings – size, type, brand?

Added.

L147-148: There is little consensus on the definition and markers of juvenile vs subadult bats, yet these stages are likely very different in their susceptibility to disease. The authors should be more specific here on how they determined these to be subadults and not juveniles (e.g., define that a subadult is a volant non-adult). I wonder whether they would be better to refer to non-adults as simply “immature”, though, to avoid confusion when comparing across studies.

We acknowledge the reviewer's observation regarding the absence of a consensus on age categorization criteria and terminology. In response to this valuable feedback, we have expanded our definition on that, specifically noting that subadults are individuals that have achieved volancy but are not yet adults, in the Material and Methods section (L152). We appreciate the suggestion to consider using the term "immature" as an alternative to "subadult" in our manuscript. However, we believe that "immature" may be overly encompassing, potentially implying a broader range of developmental stages, including non-volant early-developmental juveniles. It's important to note that our capture method relied on mist nets across cave entrances, which primarily captured volant individuals at a specific stage of development. Therefore, we have chosen to retain the term "subadult" to accurately reflect the specific developmental stage under consideration.

L151: More information needed on the ethanol – what % ethanol?

We used 90% ethanol in our study to preserve the samples and included this specific information (L155) in the Materials and Methods section to provide clarity on our sample preservation methods.

L182: More information needed on the RNA copies/ul – how was this viral load quantified (e.g., droplet digital PCR?) and was this quantified in the same laboratory, and therefore directly comparable to the Ct value, or taken from another paper? Ct values alone are not directly comparable across laboratories. This could be specified at first mention, e.g., “...equivalent to >15 CoV-182 RNA copies/μL in our assays.” This information is not currently provided in the Supplementary Material.

Thank you for raising a critical point regarding the quantification of viral load. We agree that Ct values can't be compared between different studies or papers. However, in our study, we used photometrically quantified *in vitro* transcribed RNA as positive controls and calibrator to control for run-to-run consistency. Furthermore, all data was generated within the same laboratory and using consistent methods, making them comparable. We made the necessary adjustments to clarify the corresponding section in the Supplementary Material (L177-193).

L200: The word “rate” has a specific meaning (is a quotient of two quantities in different units), and I would caution the authors to be intentional in its use. “Shedding rate” here implies shedding over

time, but the data in question are singular Ct values. I suggest replacing all mention of “shedding rate” with “viral load” (but be sure to establish that Ct is a proxy for viral load).

We followed suit and exchanged “shedding rate” against “viral load” (please see L204-205, L271, and L351).

L208-213: I understand now that the response variable is at the individual level, while the explanatory variables are at the sampling level (i.e., unique site/time). However, given there are no explanatory variables at the individual level, why don’t the authors use CoV infection prevalence per site/time combination, instead of CoV infection probability per individual? My concern is that the sample size is artificially inflated – for example, if there were 500 bats from a single site/time combination, the model will fit the exact same values for explanatory variables 500 times. Unless I have misunderstood the structure of the model?

Indeed, we aimed to investigate the factors that determine the risk of an individual bat for a CoV infection while considering the influence of its co-roosting companions. The explanatory variables in our model represent characteristics specific to each unique site and time combination, as these factors vary between different sampling locations and time points. To address the issue of repeating values within each site/time combination, we employed a generalised mixed effect model. This approach allowed us to distinguish between fixed and random effects. By incorporating the mixed effects, we effectively accounted for the nested nature of the variables and controlled for the artificial inflation of sample size. In summary, our decision to model CoV infection probability per individual while incorporating explanatory variables at the site and time level was driven by our interest in examining how group-level characteristics influence the risk of individual infection, rather than focusing solely on the prevalence as the end outcome.

L236-264: “...and subadults...” – is this ALL subadults, or subadults of the named species?

Thank you for the clarification. We have adjusted the sentence to correctly indicate that it refers to all subadults, irrespective of the named species.

L295-296: “...the best models incorporating Shannon diversity explained... and retained Shannon diversity...” – were these models presented specifically because they retained Shannon diversity? If so, the fact that they retained Shannon diversity isn’t a result (i.e., doesn’t need to be repeated the second time).

Correct, the mention of retaining Shannon diversity was not necessary and has caused redundancy. We improved the conciseness of our manuscript by deleting this part and avoiding such repetition. Thank you for bringing this to our attention.

L310: remove “the” in front of “species communities”

Removed.

L329-361: I found the shift to “chiropterans” from just “bats” slightly odd.

We made this adjustment to introduce a broader taxonomic perspective and to encompass all members of the Chiroptera order, which includes not only bats but also e.g. flying foxes. However, we understand that this change might have felt slightly unusual, and we will review the text to ensure that the terminology flows smoothly and is consistent throughout the manuscript. Thank you for bringing this to our attention.

346-357: Confusing structure with “though” and “while” in the same sentence, and with the placement of the citation. Also “co-occur” and “together” mean the same thing, so one of these words is redundant. Suggest: “We had previously found only beta-CoVs 2b and 2bBasal to co-occur frequently [citation], but here found that co-infections of beta-CoV 2b and alpha-CoV 229E-like are less likely.”

Thank you for the suggestion. We've revised the sentence structure as per your recommendation.

L349-350: “...More than 40% of locals...” this study didn’t survey ALL locals. Its 40% of survey respondents, which could be a biased representation of locals. Suggest “...40% of local survey respondents...”

Correct, it's more accurate to specify that it's 40% of local survey respondents. We have specified this in the sentence.

L352-353: Not all bat species

We modified the sentence by including "certain" before "bat species" to signify that the reaction to habitat changes is not uniform across all bat species.

L354-361: The authors haven’t analysed the association with disturbance in this manuscript, and looking at the supplementary information, it seems that the disturbance to the sampled caves was comparable (i.e., there was no pristine, or minimally impacted cave to compare with). Be careful not to imply these relationships in this section.

It is important to note that the disturbance types varied across the sampled caves, encompassing a range of different factors. While there were differences in disturbance types and grades between the caves, it is true that there was no pristine or minimally impacted cave available for direct comparison. This is correct - we did not directly analyse the association with disturbance in this manuscript, and we understand the importance of clarity in our statements. We will ensure that our revised manuscript does not imply such relationships in the relevant sections. Thank you for pointing out this issue.

L363 & throughout – be careful using the word “disease” when referring to bat pathogens. The pathogens in this paper don’t cause disease to bats

We appreciate the reviewer's caution regarding the use of the term "disease" when referring to bat pathogens. While we fully agree with the caution that needs to be taken when communicating about bats and understand the importance of precision in language, we faced a challenge of navigating fixed terminology such as "diversity disease relationship" or "disease dynamics" (e.g. infection dynamics would refer more to the course of an infection) in the context of our study. We have made efforts to avoid using the term "disease" when describing CoV infections in bats, but when common terminology in the literature includes "disease", we have maintained it to prevent potential confusion.

389: host[s]

Corrected.

391: instance, not instances

Changed.

L407: Similar to a previous comment, I would consider referring to “immature individuals” instead of “juveniles”

We have replaced juveniles with immature individuals in that sentence, providing a more precise and comprehensive description.

Supplementary Table 2 isn't interpretable without referring to the cited manuscripts. It would be worth elaborating on how categories were assigned, especially to explain Biotic Vulnerability Scores vs Biotic Vulnerability Index (e.g., the key on L66-70 states that a score between 1-1.99 = index A, yet the table shows that caves with scores of 1.6 and 1.7 have an index of B), and to explain the Bat Cave Vulnerability Index.

We have now provided a more detailed description of the categorisation process and the Bat Cave Vulnerability Index in the Supplementary Material (L63-85). We have clarified the assignment of Biotic Vulnerability Scores and their corresponding index categories. This additional information should make Supplementary Table 2 more interpretable and provide a clearer understanding of how the scoring and categorization were conducted.

Supplementary Table 7: the number of individuals and subadults should be presented without a decimal place

Done.

Supplementary Table 12: include the proportion of variation explained by the model in this table.

Thank you for your suggestion. We'd like to clarify that in Supplementary Table 12, the model weight already represents the proportion of variation explained by the model. This metric is a common way to assess the model's explanatory power.

REVIEWERS' COMMENTS

Reviewer #2 (Remarks to the Author):

General comments

The authors attended to all comments and suggestions. The manuscript now is focused on the causal connections between competent host species and the abundance of subadults, with CoV prevalence. The Dilution Effect hypothesis is raised, but duly relativized, mainly because it is not possible to disentangle the effects of changes in host abundance from intrinsic

properties of biodiversity. Shifts in species assemblages do not necessarily always occur in the same direction, and this is properly discussed in the manuscript.

In relation to the discussion of the nature of the *H. caffer* complex, the authors have provided evidence about the distinction between its lineages. Although I recommended an additional analysis treating them as a single host, I consider the authors have made a rational defense of their decision, especially by the fact that these lineages are genetically isolated, and because they have different levels of competence as hosts.

Minor comment

From the rebuttal letter:

“As the reviewer has recognised, we moved away from emphasising species diversity as explanatory factor for CoV” ... “and specifically emphasize that the difference in CoV is linked to changes in the relative abundance of competent hosts”

But in the Abstract:

“Broadly, bat species varied in CoV competence, and highly competent species were dominant in less diverse communities, leading to increased CoV prevalence in less diverse bat assemblages”

The phrase remains tendentious. The authors continue to highlight the connection between CoV prevalence and bat diversity, which should be avoided. Instead, they should assert that

it is the relative abundance of competent species (and of subadults) rather than diversity, that accounts for CoV prevalence.

Reviewer #4 (Remarks to the Author):

This is the third time I have reviewed this manuscript. The major concern of my previous review centered around the model selection process used by the authors. The author's current response and edits have now more clearly described and represented their statistical methods, and I appreciate the author's diligence in preparing such a thorough response. I especially like the addition of the bounding outlines in Supplementary Table 12 to show candidate model sets.

What the authors have presented isn't strictly how model selection should work, but I recognize that picking one model from a competitive set of models isn't uncommon in ecology. I appreciate that this selection is based on ecological understanding, but all candidate models should be ecologically reasonable. If it's true that the model averaging came out with comparable results, this statistical distinction may not change the ecological interpretation. The authors haven't presented enough in the main text or SI to confirm this, though. One would have to run the script on GitHub to check.

Overall, the authors have done a great job incorporating reviewer comments, and the manuscript is much clearer. I would still encourage the authors to interpret the model averaged output for each candidate set in the main text, rather than the full model from each set, or present the model averaging results in the SI, however I will leave this decision to the editor and/or authors.

If it is helpful, I have provided a more thorough explanation of model averaging below. My intention with this is to be helpful, not condescending! Apologies if this is information the authors already know and understand.

Additional Minor Comments:

Supplementary Table 12 – Consider including the full set of models and bold models with $\Delta AICC \leq 2.00$. Model weights can be used to estimate the relative importance of variables under consideration, done by summing the Akaike weights for each model in which that variable appears. E.g., if a particular predictor appears in all of the top models, then its summed Akaike weight will tend towards 1. The summed weights can be used to rank the various predictors in terms of importance, and could provide further support to the authors interpretation on the importance of each variable.

Caption in Supplementary Table 12 – In the caption the authors state “... and weights representing the variation explained by each model”. One might confuse this description with the model R2 and should be edited. I'm assuming this is Akaike weight and not R2, given the context of model averaging (if the full set was presented one could check that the weights add to 1). The Akaike weight for a given model is a value between 0 and 1, and can be considered as analogous to the probability that a given model is the best approximating model. So, in the top model set of Supplementary Table 12, you can see that the full model (bolded) has a weight of 0.182, which can be interpreted as meaning that there is 18.2% chance that it really is the best approximating model describing the data given the candidate set of models considered.

More Information on Model Averaging

The authors have sets of models, for which they've used AIC model selection to evaluate the best model. Each model set can be visualized in Supplementary Table 12 (each model set is framed).

The authors have accurately stated that models falling within $\Delta AICC \leq 2.00$ are competitive within each set. Typically, if there are multiple models in a candidate set with $\Delta AICC \leq 2.00$ one would not make inferences from any single model as it could be misleading.

Supplementary Table 12 shows that the authors often have multiple competitive models.

The authors have now also provided the Akaike weight per model in the candidate set

(Supplementary Table 12). The Akaike weight for a given model is considered as analogous to the probability that a given model is the best approximating model. So, in the top model set of Supplementary Table 12, one can see that the full model (bolded) has a weight of 0.182, which can be interpreted as meaning that there is 18.2% chance that it really is the best approximating model describing the data given the candidate set of models considered. This is quite low, but because it is the full model for this candidate set, this is the model the authors would interpret in the main text.

Because there is model uncertainty (based on the $\Delta AICC$ and Akaike weight), one would typically use model averaging across the full set of models. This produces parameter and error estimates that are not conditional on any one model but instead derive from weighted averages of these values across multiple models. The authors say in the rebuttal that they have done this, so I don't understand why they wouldn't present it (in the main text or SI). Their reasoning for presenting the full models was to allow comparison across model sets, but if they applied model averaging to all model sets in the same way, they should be just as comparable.

Symonds and Moussalli (2011) 'A brief guide to model selection, multimodel inference and model averaging in behavioural ecology using Akaike's information criterion' is a great resources for more information (DOI 10.1007/s00265-010-1037-6).

REVIEWERS' COMMENTS

Reviewer #2 (Remarks to the Author):

General comments

The authors attended to all comments and suggestions. The manuscript now is focused on the causal connections between competent host species and the abundance of subadults, with CoV prevalence. The Dilution Effect hypothesis is raised, but duly relativized, mainly because it is not possible to disentangle the effects of changes in host abundance from intrinsic properties of biodiversity. Shifts in species assemblages do not necessarily always occur in the same direction, and this is properly discussed in the manuscript.

In relation to the discussion of the nature of the *H. caffer* complex, the authors have provided evidence about the distinction between its lineages. Although I recommended an additional analysis treating them as a single host, I consider the authors have made a rational defence of their decision, especially by the fact that these lineages are genetically isolated, and because they have different levels of competence as hosts.

We thank the reviewer for the feedback and acknowledging our efforts in addressing the comments. We concur that there is substantial evidence supporting the distinction between the lineages of the *Hipposideros caffer* complex. As stated in our prior responses, lumping the lineages back together is ignoring intrinsic differences in ecology, (immune-) genetic divergence and competence to CoV infections. We appreciate the understanding of our rationale in this regard.

Minor comment

From the rebuttal letter:

“As the reviewer has recognised, we moved away from emphasising species diversity as explanatory factor for CoV”... “and specifically emphasize that the difference in CoV is linked to changes in the relative abundance of competent hosts”

But in the Abstract:

“Broadly, bat species varied in CoV competence, and highly competent species were dominant in less diverse communities, leading to increased CoV prevalence in less diverse bat assemblages”. The phrase remains tendentious. The authors continue to highlight the connection between CoV prevalence and bat diversity, which should be avoided. Instead, they should assert that it is the relative abundance of competent species (and of subadults) rather than diversity, that accounts for CoV prevalence.

We appreciate the reviewer's input and understand the concern expressed. While we prefer not to delve into semantic debates, it's crucial to highlight that the sentence immediately preceding the one in question explicitly states, "Prevalence and infection likelihood for both phylogenetically distinct CoVs was influenced by the abundance of competent species and naïve subadults." This underscores the importance of the relative abundance of competent hosts in influencing CoV prevalence. We promptly offer context for this statement, emphasizing the central role of competent hosts. Moreover, this nuanced understanding is further elaborated upon in the discussion section of the manuscript. Remaining committed to accurately representing our research findings, we are confident that the current presentation effectively communicates that the relationship between diversity and disease is not a direct link but rather influenced by host characteristics.

Reviewer #4 (Remarks to the Author):

This is the third time I have reviewed this manuscript. The major concern of my previous review centered around the model selection process used by the authors. The author's

current response and edits have now more clearly described and represented their statistical methods, and I appreciate the author's diligence in preparing such a thorough response. I especially like the addition of the bounding outlines in Supplementary Table 12 to show candidate model sets.

What the authors have presented isn't strictly how model selection should work, but I recognize that picking one model from a competitive set of models isn't uncommon in ecology. I appreciate that this selection is based on ecological understanding, but all candidate models should be ecologically reasonable. If it's true that the model averaging came out with comparable results, this statistical distinction may not change the ecological interpretation. The authors haven't presented enough in the main text or SI to confirm this, though. One would have to run the script on GitHub to check.

Overall, the authors have done a great job incorporating reviewer comments, and the manuscript is much clearer. I would still encourage the authors to interpret the model averaged output for each candidate set in the main text, rather than the full model from each set, or present the model averaging results in the SI, however I will leave this decision to the editor and/or authors.

As per the request, we have incorporated the outcomes derived from model averaging in the SI (Tables 16-18). This now demonstrates that selection did not influence the ecological interpretation. The model outcomes initially showcased in the main text remain unchanged, as they were the sole models with $\Delta AICC \leq 2.00$, rendering model averaging inapplicable. Supplementary Table 12 is retained for the sake of clarity about which models were competitive.

If it is helpful, I have provided a more thorough explanation of model averaging below. My intention with this is to be helpful, not condescending! Apologies if this is information the authors already know and understand.

Thank you for that. We did not take it as condescending but rather constructive, albeit the information is known to us.

Additional Minor Comments:

Supplementary Table 12 – Consider including the full set of models and bold models with $\Delta AICC \leq 2.00$. Model weights can be used to estimate the relative importance of variables under consideration, done by summing the Akaike weights for each model in which that variable appears. E.g., if a particular predictor appears in all of the top models, then its summed Akaike weight will tend towards 1. The summed weights can be used to rank the various predictors in terms of importance, and could provide further support to the authors interpretation on the importance of each variable.

We thank the reviewer for this suggestion, yet we feel confident that readers can easily derive this information from Supplementary Table 12 and the models outlined in the main text.

Caption in Supplementary Table 12 – In the caption the authors state "... and weights representing the variation explained by each model". One might confuse this description with the model R2 and should be edited. I'm assuming this is Akaike weight and not R2, given the context of model averaging (if the full set was presented one could check that the weights add to 1). The Akaike weight for a given model is a value between 0 and 1, and can be considered as analogous to the probability that a given model is the best approximating model. So, in the top model set of Supplementary Table 12, you can see that the full model (bolded) has a weight of 0.182, which can be

interpreted as meaning that there is 18.2% chance that it really is the best approximating model describing the data given the candidate set of models considered.

We changed the caption in accordance with the recommendation from the reviewer.

More Information on Model Averaging

The authors have sets of models, for which they've used AIC model selection to evaluate the best model. Each model set can be visualized in Supplementary Table 12 (each model set is framed).

The authors have accurately stated that models falling within $\Delta\text{AICC} \leq 2.00$ are competitive within each set. Typically, if there are multiple models in a candidate set with $\Delta\text{AICC} \leq 2.00$ one would not make inferences from any single model as it could be misleading. Supplementary Table 12 shows that the authors often have multiple competitive models.

The authors have now also provided the Akaike weight per model in the candidate set (Supplementary Table 12). The Akaike weight for a given model is considered as analogous to the probability that a given model is the best approximating model. So, in the top model set of Supplementary Table 12, one can see that the full model (bolded) has a weight of 0.182, which can be interpreted as meaning that there is 18.2% chance that it really is the best approximating model describing the data given the candidate set of models considered. This is quite low, but because it is the full model for this candidate set, this is the model the authors would interpret in the main text.

Because there is model uncertainty (based on the ΔAICC and Akaike weight), one would typically use model averaging across the full set of models. This produces parameter and error estimates that are not conditional on any one model but instead derive from weighted averages of these values across multiple models. The authors say in the rebuttal that they have done this, so I don't understand why they wouldn't present it (in the main text or SI). Their reasoning for presenting the full models was to allow comparison across model sets, but if they applied model averaging to all model sets in the same way, they should be just as comparable.

Symonds and Moussalli (2011) 'A brief guide to model selection, multimodel inference and model averaging in behavioural ecology using Akaike's information criterion' is a great resources for more information (DOI 10.1007/s00265-010-1037-6).

Thank you for providing this concise recap. While we were aware of the theory, your summary was greatly appreciated. We extend our gratitude for your continued support throughout the three revision rounds of our manuscript.